# MASTER-NAADP: a membrane permeable precursor of the Ca$^{2+}$ mobilizing second messenger NAADP

Sarah Krukenberg[1,10], Franziska Möckl[2,10], Mariella Weiß[2], Patrick Dekiert[1], Melanie Hofmann[1], Fynn Gerlach[2], Kai J. Winterberg [2], Dejan Kovacevic[2], Imrankhan Khansahib[2], Berit Troost[2], Macarena Hinrichs[2], Viviana Granato [2], Mikolaj Nawrocki[3], Tobis Hub[4,5], Volodymyr Tsvilovskyy[4,5], Rebekka Medert [4,5], Lena-Marie Woelk[6], Fritz Förster[6], Huan Li [7,8], René Werner [6], Marcus Altfeld[9], Samuel Huber [3], Oliver Biggs Clarke [7,8], Marc Freichel [4,5], Björn-Philipp Diercks [2], Chris Meier[1,11] & Andreas H. Guse [2,11] ✉

Upon stimulation of membrane receptors, nicotinic acid adenine dinucleotide phosphate (NAADP) is formed as second messenger within seconds and evokes Ca$^{2+}$ signaling in many different cell types. Here, to directly stimulate NAADP signaling, MASTER-NAADP, a Membrane permeAble, STabilized, bio-rEversibly pRotected precursor of NAADP is synthesized and release of its active NAADP mimetic, benzoic acid C-nucleoside, 2'-phospho-3'F-adenosine-diphosphate, by esterase digestion is confirmed. In the presence of NAADP receptor HN1L/JPT2 (hematological and neurological expressed 1-like protein, HN1L, also known as Jupiter microtubule-associated homolog 2, JPT2), this active NAADP mimetic releases Ca$^{2+}$ and increases the open probability of type 1 ryanodine receptor. When added to intact cells, MASTER-NAADP initially evokes single local Ca$^{2+}$ signals of low amplitude. Subsequently, also global Ca$^{2+}$ signaling is observed in T cells, natural killer cells, and Neuro2A cells. In contrast, control compound MASTER-NADP does not stimulate Ca$^{2+}$ signaling. Likewise, in cells devoid of HN1L/JPT2, MASTER-NAADP does not affect Ca$^{2+}$ signaling, confirming that the product released from MASTER-NAADP is a bona fide NAADP mimetic.

Nicotinic acid adenine dinucleotide phosphate (NAADP) was discovered as an impurity of commercial NADP preparations[1] and turned out to be the most potent Ca$^{2+}$ mobilizing 2nd messenger known to date (reviewed in refs. 2–5). NAADP is formed rapidly within seconds upon stimulation of plasma membrane receptors[6–8]. Currently, two major models for formation of NAADP exist: either NAADP is formed by the 'base-exchange' reaction through either CD38[9] or SARM1[10], or via oxidation of its reduced derivative NAADPH[11]. The base-exchange

[1]Organic Chemistry, University of Hamburg, 20146 Hamburg, Germany. [2]The Calcium Signalling Group, Department of Biochemistry and Molecular Cell Biology, University Medical Center Hamburg-Eppendorf, 20246 Hamburg, Germany. [3]Section of Molecular Immunology and Gastroenterology, I. Department of Medicine, University Medical Center Hamburg-Eppendorf, 20246 Hamburg, Germany. [4]Institute of Pharmacology, Heidelberg University, Heidelberg, Germany. [5]DZHK (German Centre for Cardiovascular Research), partner site Heidelberg/Mannheim, Heidelberg, Germany. [6]Department of Applied Medical Informatics, University Medical Center Hamburg-Eppendorf, 20246 Hamburg, Germany. [7]Department of Anesthesiology, Columbia University Irving Medical Center, New York, NY, USA. [8]Department of Physiology and Cellular Biophysics, Columbia University, New York, NY, USA. [9]Department of Immunology, University Medical Center Hamburg-Eppendorf, 20246 Hamburg, Germany. [10]These authors contributed equally: Sarah Krukenberg, Franziska Möckl. [11]These authors jointly supervised this work: Chris Meier, Andreas H. Guse. ✉e-mail: guse@uke.de

reaction catalyzed by CD38 requires special conditions, e.g. acidic pH and excess of nicotinic acid[9], although it was recently reported that SARM1 may catalyze the same reaction also at neutral pH[12]. Further, the acidic pH present in the endo-lysosomal lumen allows for NAADP formation by CD38 using NADP and NAAD as substrates to generate NAADP via the base-exchange reaction. Solid evidence for this endo-lysosomal model of NAADP production was obtained in lymphokine activated killer cells stimulated by interleukin-8[13]. Recently, NAADPH oxidation by plasma membrane spanning NADPH oxidases was demonstrated as alternative pathway of NAADP generation; in contrast to the endo-lysosomal model, this process proceeds at neutral pH in the cytosol just below the internal leaflet of the plasma membrane and requires both NAADPH and $O_2$ as substrates[11].

The $Ca^{2+}$ release mechanism operated by NAADP requires NAADP receptor/NAADP binding proteins, as demonstrated by Lin-Moshier et al.[14] and Walseth et al.[15] Based on these results the unifying hypothesis that links NAADP to activation of $Ca^{2+}$ channels was published[16]. Two such NAADP receptors/NAADP binding proteins were recently identified as HN1L/JPT2[17,18] or Lsm12[19]. These NAADP receptors/NAAP binding proteins activate different ion channels, e.g. type 1 ryanodine receptor (RYR1)[17], two-pore channel 1 (TPC1)[18], or TPC2[19], resulting in release of $Ca^{2+}$ either from endoplasmic reticulum (ER) through RYR1 or from endo-lysosomes through TPC1 and/or TPC2. Primary release of $Ca^{2+}$ via these NAADP-directed mechanisms results in local $Ca^{2+}$ release, e.g. $Ca^{2+}$ microdomains in T cells[20,21]. However, very rapid amplification mechanisms enlarge the initial local signals finally resulting in global $Ca^{2+}$ signaling; among these amplification mechanisms are (i) $Ca^{2+}$ induced $Ca^{2+}$ release through RYR[22], (ii) enhancement of $IP_3$ evoked $Ca^{2+}$ release through $IP_3$Rs[23], (iii) store-operated $Ca^{2+}$ entry (SOCE)[21], or amplification via purinergic signaling involving purinergic P2X4 and P2X7[24].

To study NAADP signaling, a couple of chemical biology and pharmacological tools have been developed. Ned-19[25] and BZ194[26] were both introduced in 2009 as NAADP antagonists and have been used to block signaling of endogenously formed NAADP in different cell types. The known $Ca^{2+}$ mobilizing second messengers, including NAADP, are charged at physiological pH and thus are not membrane-permeant. Therefore, to study effects of NAADP directly in intact cells, microinjection[20] or infusion via patch-clamp pipette[27] must be carried out. However, these methods are laborious and have disadvantages, e.g. the risk to contaminate the injected volume by traces of extracellular buffer containing high $Ca^{2+}$ concentration during microinjection, or dilution of relevant cytosolic proteins into the patch-clamp pipette in whole-cell mode measurements. Thus, NAADP-acetoxymethyl ester (NAADP-AM) was synthesized and characterized as membrane permeant precursor for NAADP[28,29]. NAADP-AM was, among other cell types, successfully used to evoke global $Ca^{2+}$ signals in cytotoxic T cells[30].

Since it is known that acetoxymethylation of pyrophosphate moieties in small molecules such as dinucelotides may result in unstable products[31], here we develop a Membrane permeAble, STabilized, bio-rEversibly pRotected precursor of NAADP (MASTER-NAADP). Synthesis and full chemical characterization of MASTER-NAADP and a similar lipophilic precursor for the inactive NADP, MASTER-NADP, are described. Further, MASTER-NAADP and MASTER-NADP are evaluated regarding their ability to evoke local and/or global $Ca^{2+}$ signaling in T cells, natural killer cells, and a neuronal cell line. In addition, the deprotected MASTER compounds are directly characterized regarding biological activity in permeabilized T cells and by lipid planar bilayer measurements using RYR1.

## Results

### Synthesis of MASTER-NAADP

First, the "southern" adenosine-2′,5′-phosphate derivative was prepared. Xylose 1 (Fig. 1) was first protected in the 1,2-position with an isopropylidene group (99%). In solution xylose 1 exists in the thermodynamically favored pyranose form over the furanose form. This equilibrium can be shifted if the sugar is forced into the furanose form by the introduction of the isopropylidene protecting group[32]. The 5′-protecting group was introduced following standard reaction conditions using benzoyl chloride[33]. To obtain the fluorinated compound 5, the isopropylidene group was cleaved and the 1′-position was protected with a methoxy group (86%). Next, the 3′-position was fluorinated with diethylaminosulfur trifluoride (DAST). Instead of using 6.0 eq. as described before[34], the fluorination was carried out by adding only 1.5 eq. of DAST because using more than 1.5 eq. did not result in increased product formation. Acetyl protecting groups were introduced according to literature procedures to obtain the desired ß-configurated product 7[35] in the following step. The N-glycosylation was carried out according to the Vorbrüggen method[36] following by ester cleavage of the protecting groups under basic conditions, yielding compound 8. After the 5′-position was tert-butyldimethylsilyl (TBDMS)-protected, the bisacyloxybenzyl (AB)-masked phosphate group was introduced into the 2′-position. Therefore, the bis(AB)-phosphoramidite 12 was synthesized starting from 4-hydroxyl benzyl alcohol 10 which was first converted into acyloxybenzyl alcohol 11. Phosphoramidite 12 was prepared using the alcohol 11 and subsequently reacted with the 2′-position using acid-catalysis[37] to give the silyl-protected 2′-phosphate 13 in a good yield of 87%[38]. After cleavage of the TBDMS group with triethylamine-trihydrofluoride to give the 2′-phosphate 14, the monophosphorylation was carried out by the Yoshikawa phosphorylation method to yield the 2′,5′-diphosphate 15[39].

The "northern" C-nucleoside 19 was synthesized according to Krohn et al.[40] (Fig. 2). The Yamaguchi esterification of the carboxylic acid 19[41] was carried out using the carbonate mask 22. After complete debenzylation, the monophosphorylation was carried out at the 5′-position of the carbonate-masked C-nucleoside 24 (42%)[39]. MASTER-NAADP 26 was successfully obtained in a yield of 57 % by coupling the two monophosphates 25 and 15[42].

### Synthesis of MASTER-NADP

The synthesis of the amide comprising C-nucleoside monophosphate 32 ("northern" part) follows the synthesis route of Supplementary Fig. 1. In contrast to the esterification step (Fig. 2A), the amide 30 is synthesized using HBTU[43].

Supplementary Fig. 2 summarizes the synthesis of the "southern" building block 40 and the NADP-derivative 42. The 5′- and 3′-hydroxyl group of adenosine 35 were protected with the tetra-iso-propyldisiloxan-1,3-diyl (TIPDS) group[44] to allow selective phosphoramidite coupling at the 2′-position to obtain compound 38[38]. The synthesis of the masking moiety is analogous to the phosphoramidite protocol in Fig. 1[37]. To prevent 2′- to 3′-phosphate group migration and to allow subsequent coupling to the NADP-derivative 40, the 5′-position was selectively deprotected[45]. Direct monophosphorylation of the 5′-position was not possible because a competing reaction to the silanol phosphate was observed. Therefore, the 5′-H-phosphonate (39; 89%) was synthesized first, which was oxidized to give the monophosphate 40 in a yield of 99%[46]. By coupling the two monophosphates 32 and 40, the protected MASTER-NADP 41 was obtained in a yield of 64% (Supplementary Fig. 2)[42]. To obtain the NADP-derivative 42, the silyl protecting group at the 3′-position was cleaved off using tris(dimethylamino)sulfonium difluorotrimethylsilicate (TASF)[47].

### Deprotected MASTER-NAADP releases $Ca^{2+}$ from permeabilized cells and activates RYR1 channel opening in lipid planar bilayers

The dinucleoside diphosphate liberated by endogenous esterase from MASTER-NAADP is a very close derivative of NAADP, benzoic acid C-nucleoside, 2′-phospho-3′F-adenosine-diphosphate. In fact, the major product of in vitro digestion of MASTER-NAADP by porcine liver esterases was purified by HPLC and identified by mass spectrometry as

**Fig. 1 | Synthesis of MASTER-NAADP and derivatives I Reagents and conditions.**
**[a]** (i) 0.10 eq. CuSO$_4$, 2.8 eq. conc. H$_2$SO$_4$, acetone, 3 h, rt, (ii) HCl-solution (0.2 %), 6 h, rt, 99 %; **[b]** 1.5 eq. Et$_3$N, 1.1 eq. BzCl, CH$_2$Cl$_2$, 1 h, 0 °C, 92 %; **[c]** 0.2 eq. I$_2$, MeOH, 4 h, 70 °C, 86 %; **[d]** 1.5 eq. DAST, dry CH$_2$Cl$_2$, 20 h, -78 °C → rt, 43 %; **[e]** 4.7 eq. Ac$_2$O, 4.7 eq. conc. H$_2$SO$_4$, AcOH, 19 h, rt, 93 %; **[f]** (i) 1.5 eq. *N*-benzoyladenine, 1.2 eq. BSA, C$_2$H$_4$Cl$_2$, 1 h, 90 °C, (ii) 1.0 eq. **6**, 4.0 eq. TMSOTf, C$_2$H$_4$Cl$_2$, 5.5 h, 90 °C, 61 %; **[g]** ammonia 7 N in MeOH, 28 h, 70 °C, 84 %; **[h]** 1.5 eq. TBDMSCl, dry pyridine, 24 h,

rt, 71 %; **[i]** 1.2 eq. 4-hydroxyl benzyl alcohol **10**, 1.2 eq. Et$_3$N, 1.0 eq. heptanoyl chloride, THF, 2 h, 0 °C, 87 %; **[j]** 1.0 eq. dichloro-*N,N*-di*iso*propylaminophosphoramidite, 2.2 eq. **11**, 2.3 eq. Et$_3$N, THF, 0 °C → rt, 16 h, 92 %; **[k]** (i) 1.3 eq. **12**, 1.2 eq. pyridinium trifluoracetate, CH$_2$Cl$_2$, 50 min, rt, (ii) 1.5 eq. *tert*-butylhydroperoxide (5.0 M in decane), 0 °C, 1 h, 87 %; **[l]** 5.2 eq. triethylamine-trihydrofluoride, CH$_2$Cl$_2$, 19 h, rt, 81 %; **[m]** 4.0 eq. POCl$_3$, TMP, 6.5 h, 0 °C, 31 %.

benzoic acid C-nucleoside, 2′-phospho-3′F-adenosine-diphosphate (deMASTER-NAADP; Fig. 2B). The NADP derivative released from MASTER-NADP was also obtained by in vitro digestion using porcine liver esterases and eluted as only partially resolved double peak that represents the 2′- and 3′-phosphate isomers of benzamide C-nucleoside, phospho-adenosine-diphosphate (deMASTER-NADP)[48,49] (Supplementary Fig. 3A). deMASTER-NADP lacks the essential 3-carboxyl group at the benzoic acid moiety (which replaces the nicotinic acid moiety here) and should not mimic NAADP's Ca$^{2+}$ releasing activity. Thus, only the expected NAADP mimic deMASTER-NAADP contains all structural moieties reported to be crucial for biological activity of NAADP, the 3-carboxyl group of nicotinic acid, the 6-amino group and the 2′-phosphate of the adenosine moiety[50]. Further, modeling of deMASTER-NAADP and its overlay with NAADP indicate very similar overall architecture of NAADP and its synthetic mimic released from MASTER-NAADP (Fig. 2C).

The deprotected MASTER compounds were then validated in two systems: (i) Ca$^{2+}$ release in saponin-permeabilzed Jurkat T cells, and (ii) current recordings in planar lipid bilayers containing RYR1 and NAADP receptor HN1L/JPT2 (Fig. 3). In the absence of exogenous addition of recombinant HN1L/JPT2 neither natural NAADP, nor deMASTER-NADP, nor deMASTER-NAADP released Ca$^{2+}$ (Fig. 3A). In contrast, in the presence of HN1L/JPT2, both NAADP and deMASTER-

NAADP released Ca$^{2+}$, whereas deMASTER-NADP or NADP did not (Fig. 3B). In line with NAADP's target Ca$^{2+}$ channel in T cells, RYR1[14,17,18,23], NAADP (Fig. 3C) or deMASTER-NAADP (Fig. 3D) evoked Ca$^{2+}$ release in a Ca$^{2+}$ dependent fashion. Further, RYR1 open probability (P$_o$) was determined in lipid planar bilayers containing highly purified RYR1 (Fig. 3E,F). Whereas 100 nM deMASTER-NAADP alone only slightly elevated P$_o$, markedly increased RYR1 channel opening was observed upon combination of deMASTER-NAADP and recombinant HN1L/JPT2 (Fig. 3E, F). In contrast, deprotected control compound deMASTER-NADP in combination with recombinant HN1L/JPT2 did not increase P$_o$ (Fig. 3F).

## MASTER-NAADP evokes Ca$^{2+}$ microdomains

Next, we compared the two lipophilic precursors MASTER-NAADP and MASTER-NADP regarding their Ca$^{2+}$ mobilizing properties in different cell types.

In many cell types, NAADP has been shown, or at least has been assumed, to be involved in initializing Ca$^{2+}$ signaling processes; in fact, increases of endogenous NAADP concentrations within few seconds upon stimulation of plasma membrane receptors were reported[6–8]. Thus, Ca$^{2+}$ microdomains were recorded as described previously[21], within approx. 60 s upon extracellular addition of MASTER-NAADP or MASTER-NADP (Fig. 4). In Jurkat WT T cells MASTER-NAADP evoked

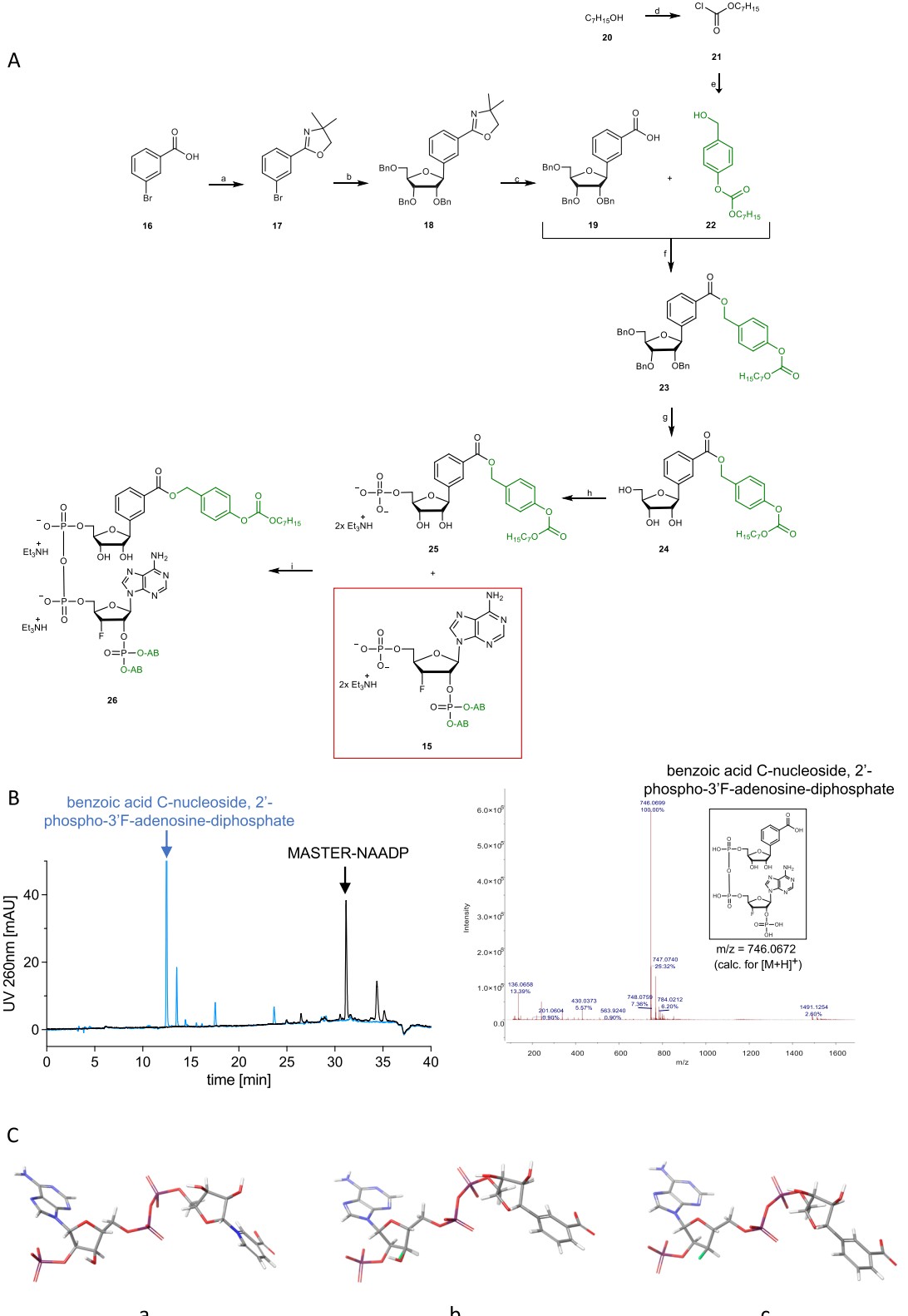

**B** benzoic acid C-nucleoside, 2'-phospho-3'F-adenosine-diphosphate

MASTER-NAADP

benzoic acid C-nucleoside, 2'-phospho-3'F-adenosine-diphosphate

m/z = 746.0672 (calc. for [M+H]⁺)

**C** a b c

Ca²⁺ microdomains that initially were observed as single low amplitude signaling events (Figs. 4A, 2.5–3.8s); within few more seconds these locally resolved Ca²⁺ signals merged into a global signal (Figs. 4A, 7.5s). In contrast, control compound MASTER-NADP did not stimulate local Ca²⁺ signals in the majority of cells (Fig. 4B). A statistically significant increase of Ca²⁺ microdomains was observed for MASTER-NAADP when compared to inactive MASTER-NADP (Fig. 4C). Further, in Jurkat T cells lacking expression of the NAADP receptor/binding protein HN1L/JPT2[17,18] only background signals for both MASTER-NAADP or MASTER-NADP were recorded (Fig. 4C; Supplementary Fig. 4A, B); similar background signals were observed upon addition of vehicle DMSO (Fig. 4C). Quantitative evaluation of the spatio-temporal pattern of Ca²⁺ microdomains was obtained by averaging Ca²⁺ microdomain data in spatial segments of the cell resembling a dart board, originating from single cells stimulated under identical conditions, as described in ref. 51. For WT Jurkat T cells, a clear difference is observed between

**Fig. 2 | Synthesis of MASTER-NAADP and derivatives II A) Reagents and conditions.** [a] Thionylchloride, kat. DMF, 2 h, 80 °C, then 2.0 eq. amino-2-methyl-1-propanol, CH₂Cl₂, 2 h, 0 °C → rt, thionylchloride, 16 h, rt, 87 %; [b] 1.5 eq. **17**, 1.6 eq. n-BuLi (1.6 M in hexane), THF, 1 h, −78 °C, then 1.0 eq. 2,3,5-tri-*O*-benzyl-ribono-lactone, THF, 2 h, −78 °C → −30 °C, 3.0 eq. triethylsilane, 2.5 eq. BF₃*Et₂O, CH₂Cl₂, 16 h,−78 °C → rt, 41 %; [c] CH₃NO₂/MeI (2:1) 16 h, reflux, then MeOH/KOH 20 %, 1:1, 16 h, reflux, quantitatively; [d] 0.33 eq. triphosgene, 1.0 eq. pyridine, CH₂Cl₂, 15 h, rt, 93 %; [e] 1.1 eq. 4-hydroxyl benzyl alcohol **10**, 1.0 eq. Et₃N, 1.0 eq. **21**, CH₂Cl₂, 15 min, 0 °C, then 3 h, rt, 74 %; [f] 1.0 eq. **19**, 1.1 eq. Yamaguchi-reagent, 1.2 eq. Et₃N, 1.2 eq. **22**, 0.3 eq. DMAP, THF, 3 h 70 °C, 16 h, rt, 58 %; [g] 3.5 eq. BCl₃ (1 M, CH₂Cl₂), 2 h, −78 °C, 67 %; [h] 2.0 eq. POCl₃, 2.5 eq. Bu₃N, trimethylphosphate, 16 h, 0 °C, 42 %. [i] 10.0 eq. TFAA, 16 eq. Et₃N, 1.0 eq. monophosphate **25**, CH₃CN, 10 min, 0 °C → Rt, then 6.0 eq. NMI, 10 eq. Et3N, CH₃CN, 10 min, 0 °C → rt, then 1.1 eq. **15**, CH₃CN, 3 h, rt, then TEAB-buffer 1 M, 15 min, rt, 57%. B) Release of MASTER-NAADP by porcine liver extract and mass spec characterization of main product. **B** MASTER-NAADP was analyzed by RP-HPLC for purity (black chromatogram). A representative out of 6 experiments is shown. Then, reaction products obtained upon incubation of MASTER-NAADP with porcine liver extract were separated by RP-HPLC (blue chromatogram). The main product was then identified by ESI-mass spectrum measured in positive ionization mode as benzoic acid C-nucleoside, 2′-phospho-3′F-adenosine-diphosphate. **C** Molecular modeling: (a) NAADP, (b) overlay of NAADP and benzoic acid C-nucleoside, 2′-phospho-3′F-adenosine-diphosphate, (c) benzoic acid C-nucleoside, 2′-phospho-3′F-adenosine-diphosphate.

cells treated with vehicle DMSO, MASTER-NAADP and control compound MASTER-NADP (Supplementary Fig. 5A). *Hn1l/Jpt2⁻/⁻* Jurkat T cells reacted to MASTER-NAADP in a comparable fashion as for MASTER-NADP in WT cells (Supplementary Fig. 5A). To demonstrate the specific effects of MASTER-NAADP over the first 15 s, unspecific signals evoked by MASTER-NADP were subtracted segment-wise (Supplementary Fig. 5B). Initial signals induced by MASTER-NAADP were observed in the center of the cells (Supplementary Fig. 5B, 1–3s), but rapidly spread to more peripheral parts of the cells (Supplementary Fig. 5B, 4–6s), followed by a phase of more even distribution of the Ca²⁺ microdomains throughout the cells (Supplementary Fig. 5B, 7–11s). Regarding kinetics, Ca²⁺ microdomains increased significantly over 30 s in WT Jurkat T cells upon stimulation by MASTER-NAADP, as compared to control conditions (Supplementary Fig. 6). In addition to the Jurkat T-lymphoma cell line, in naïve murine T cells, localized Ca²⁺ signals were also observed upon MASTER-NAADP addition (Fig. 4D). The number of Ca²⁺ microdomains was significantly higher as compared to MASTER-NADP addition (Fig. 4D, E), demonstrating effectiveness of MASTER-NAADP also in primary T cells. MASTER-NAADP was also assessed in KHYG-1 natural killer (NK) cells. Similar to Jurkat T cells, rapid local Ca²⁺ signals were observed (Fig. 4F); as for the other two cell types tested, the number of Ca²⁺ microdomains evoked by MASTER-NADP was significantly lower (Fig. 4G). As for Jurkat T cells (Supplementary Fig. 5), spatio-temporal analysis of Ca²⁺ microdomains using the dartboard representation was also conducted for KHYG-1 cells, demonstrating clearly elevated numbers of Ca²⁺ microdomains over the first 15 s upon MASTER-NAADP vs MASTER-NADP (Supplementary Fig. 7A). The corresponding subtraction plots demonstrate initial Ca²⁺ microdomains confined to a ring-like structure in the cytosol (Supplementary Fig. 7B, 1–4s). Then, Ca²⁺ microdomains were also observed close to the plasma membrane and in the inner part of the cell (Supplementary Fig. 7B, 5–9s), followed by a more even distribution of Ca²⁺ microdomains across the whole cell (Supplementary Fig. 7B, 10–15s).

In addition to T cells and NK cells, the neuronal cell line Neuro2A was stimulated with both MASTER-NAADP or MASTER-NADP (Fig. 4H, I). MASTER-NAADP stimulated local Ca²⁺ signals; soon after an almost global, but transient Ca²⁺ signal was observed (Fig. 4H). In T cells and NK cells, MASTER-NAADP evoked an approx. 3-fold increase of the number of Ca²⁺ microdomains over background (MASTER-NADP or DMSO). Of note, in Neuro2A cells MASTER-NAADP stimulated an approx. 10-fold increase of the number of Ca²⁺ microdomains, as compared to MASTER-NADP (Fig. 4I). *Hn1l/Jpt2⁻/⁻* Neuro2A cells neither responded to MASTER-NAADP nor to MASTER-NADP (Supplementary Fig. 4C, D), confirming the data obtained in Jurkat WT vs *Hn1l/Jpt2⁻/⁻* T cells (Fig. 4A vs Supplementary Fig. 4A). The absence of any Ca²⁺ microdomains above background in *Hn1l/Jpt2⁻/⁻* cells, together with direct effects of deprotected MASTER-NAADP shown in Fig. 3, strongly suggests that the NAADP derivative liberated from MASTER-NAADP, benzoic acid C-nucleoside, 2′-phospho-3′F-adenosine-diphosphate, is a bona fide Ca²⁺ mobilizing NAADP mimetic acting through the NAADP receptor/NAADP binding protein HN1L/JPT2[17,18], while benzamide adenine dinucleotide phosphate released from MASTER-NADP does not mimic NAADP.

## Global Ca²⁺ signaling stimulated by MASTER-NAADP

In addition to local Ca²⁺ signaling observed within a few seconds upon extracellular addition of MASTER-NAADP (Fig. 4), MASTER-NAADP also evoked global Ca²⁺ signaling in Jurkat T cells (Fig. 5). In these experiments Synta66 was added to block SOCE. Due to the lack of SOCE via Orai1 channels, global Ca²⁺ signals evoked by MASTER-NAADP were often of transient nature (Fig. 5A left). The control compound MASTER-NADP stimulated only negligible Ca²⁺ signals in a minority of cells (Fig. 5A middle). Commercially available NAADP-AM was also tested in Jurkat T cells for its Ca²⁺ mobilizing activity as these cells have been shown in many studies to utilize NAADP as initial Ca²⁺ mobilizing 2nd messenger during T cell receptor (TCR)/CD3 evoked Ca²⁺ signaling[11,17,20,21] and show Ca²⁺ signals upon microinjection of NAADP[20,26,52–54]. Upon addition of NAADP-AM under identical conditions as for MASTER-NAADP (100 nM final concentration), similar results as for control compound MASTER-NADP were obtained (Fig. 5A). As for the Ca²⁺ microdomain data in Fig. 4, also for analysis of global Ca²⁺ signals *Hn1l/Jpt2⁻/⁻* Jurkat T cells were employed; both MASTER-NAADP and MASTER-NADP evoked only few minor and transient Ca²⁺ signals in *Hn1l/Jpt2⁻/⁻* Jurkat T cells (Fig. 5B left, middle), similar to the Ca²⁺ signals observed in WT cells stimulated by the control compound MASTER-NADP (Fig. 5A middle). Similar results were obtained for NAADP-AM (Fig. 5B right). When comparing the Ca²⁺ peak amplitudes for the six different conditions, it became clear that the mean amplitudes of Ca²⁺ transients were not significantly different (Fig. 5C). However, MASTER-NAADP evoked a significantly higher number of global Ca²⁺ oscillations as compared to MASTER-NADP, while in *Hn1l/Jpt2⁻/⁻* T cells no global Ca²⁺ oscillations above background were observed (Fig. 5D). In addition, a similar situation as shown in Fig. 5D was observed for the percentage of responding cells (Fig. 5E). Further, as a more impactful parameter for the Ca²⁺ mobilizing activity, the 'mean responsiveness', was defined as product of peak number and amplitude. The mean responsiveness demonstrates that only MASTER-NAADP shows superior stimulation of Ca²⁺ signaling vs all other control conditions and also vs NAADP-AM (Fig. 5F).

Since NAADP-AM has been previously used over a wide concentration range[30,55–58], also higher concentrations of NAADP-AM were applied to Jurkat T cells, with no detectable Ca²⁺ signaling above background (vehicle DMSO) (Supplementary Fig. 8). Two different lots of NAADP-AM were used to exclude that a single lot was decomposed: lot #280181 shown as green tracings and lot #303795 shown in purple, both with similar results (Supplementary Fig. 8A). In contrast, subsequent stimulation of the same cells through TCR/CD3 showed rapid and sustained Ca²⁺ signaling (Supplementary Fig. 8A) confirming that Ca²⁺ signaling mechanisms were intact in these cells (Supplementary Fig. 8A). Further, 100 μM of MASTER-NAADP, MASTER-NADP, and NAADP-AM were added to intact Jurkat T cells (Supplementary Fig. 8B–E). While the peak amplitude was only different between MASTER-NAADP and vehicle DMSO (Supplementary Fig. 8B), the

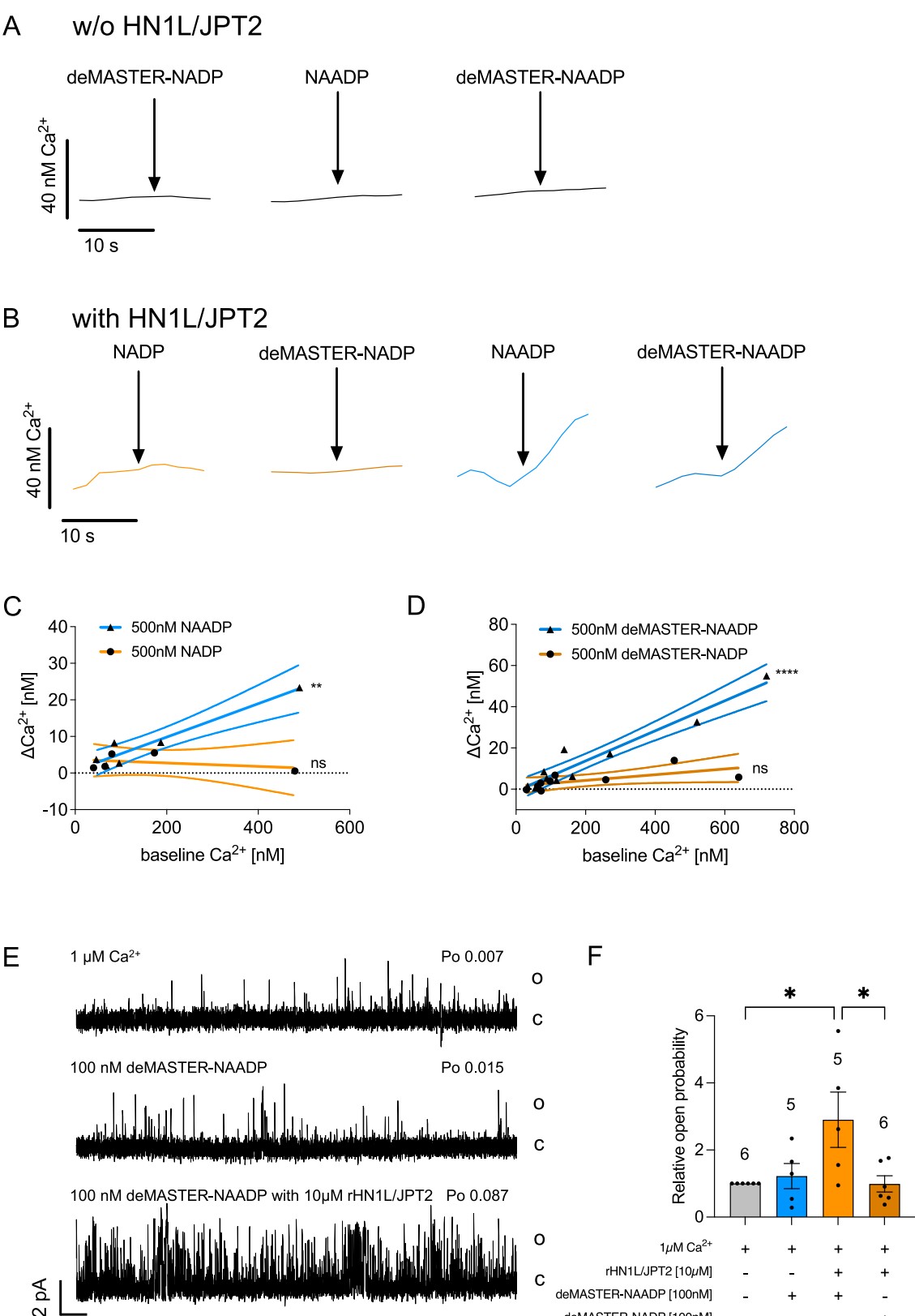

number of peaks evoked by both MASTER-NAADP or MASTER-NADP was higher as compared to NAADP-AM, but not different between MASTER-NAADP and MASTER-NADP (Supplementary Fig. 8C). At 10 μM a similar pattern among MASTER-NAADP, MASTER-NADP or NAADP-AM was observed, but at a lower level (Supplementary Fig. 8C, orange bars). Similar effects were observed for the percentage of

responding cells (Supplementary Fig. 8D). Calculation of the most robust read-out, the mean responsiveness, resulted in a somewhat improved result (Supplementary Fig. 8E). However, compared to the experiments with 100 nM MASTER-NAADP or -NADP (Fig. 5), in T cells such high concentrations of the MASTER compounds are neither necessary nor useful to study NAADP signaling.

**Fig. 3 | Deprotected MASTER-NAADP evokes Ca²⁺ signals in permeabilized Jurkat T cells and increases the open probability of RYR1.** $3.5 \times 10^7$ Jurkat T cells were permeabilized (80 µg/mL saponin) and transferred to a fluorimeter in 1 mL cuvettes with gentle magnetic stirring. Free Ca²⁺ concentration was measured every 2 s ratiometrically using emission wavelengths of CalRed (ex 488 nm, em 525/650 nm). Ca²⁺ uptake into the ER was stimulated by the addition of creatine phosphate, creatine kinase and adenosine triphosphate, followed by the addition of 15 µg recombinant human HN1L/JPT2 (except for control experiments) and 10 µM glucose 6-phosphate dehydrogenase inhibitor-1 (g6pdi-1). Deprotected MASTER-compounds or natural nucleotides were diluted in bidestilated water and injected using a Hamilton gas-tight 50µL syringe. **A** Example tracings showing no response to either biological NAADP or deprotected MASTER-compounds without the re-addition of exogenous HN1L/JPT2. Representative tracings out of 4 experiments are shown. **B** Example tracings showing increases in the free Ca²⁺ concentration after the injection of NAADP or deprotected MASTER-NAADP, but not of the respective NADP controls. A representative out of at least 5 experiments is shown. **C, D** The delta Ca²⁺ of the different experiments (mean Ca²⁺ concentration during the first 10 s after injection after subtraction of the mean Ca²⁺ concentration the last 10 s before injection) was plotted against the baseline Ca²⁺ (mean Ca²⁺ concentration the last 10 s before injection) and analyzed by linear regression with two-sided F test of the slope using GraphPad Prism 10. The graph show results from the individual experiments (symbols), linear regression (line) with 95% confidence interval (dotted lines) and significance level of the slope being non-zero, i.e. the amplitude of the responses being correlated with the basal Ca²⁺ concentration (ns = not significant, **$p < 0.01$, ****$p < 0.0001$, $N = 5$ experiments for C and $N = 8$ experiments for D). **E** Single channel analysis of deMASTER-NAADP with purified HN1L/JPT2 on RYR1. Representative single-channel current traces of purified RYR1 measured under the following conditions: upper tracing: 1 µM free Ca²⁺, middle tracing: 100 nM deMASTER-NAADP, lower tracing: 100 nM deMASTER-NAADP with 10 µM HN1L/JPT2. **F** Summarized relative open probability for each condition, presented as mean ± SEM, $n$ = biological replications as indicated above the bars. Two-sided, mixed-effects analysis with Holm-Šídák multiple comparison test, with a single pooled variance *$p < 0.05$. Each $p$ value is adjusted to account for multiple comparisons. Source data and exact $p$ values are provided as a Source Data file.

The negative results with NAADP-AM prompted us to analyze by HPLC commercially available NAADP-AM that was used in the experiments shown in Fig. 5 and Supplementary Fig. S8. Of note, NAADP-AM did not elute as a single compound, but consisted of >25 peaks of different size (Supplementary Fig. 9A), suggesting that commercially available NAADP-AM is a complex mixture of undefined individual compounds. Similar results were obtained with four different lots of NAADP-AM (Supplementary Fig. 10, all left panels), demonstrating that it was not a single decomposed lot of NAADP-AM, but appears to be a more general problem with this type of lipophilic and biocleavable NAADP derivative.

Upon digestion of NAADP-AM using porcine liver esterase (PLE), the chromatogram in Supplementary Fig. 9B showed only little changes: 2 minor peaks eluting at 23.4 min and 23.7 min disappeared (Supplementary Fig. 9B, brown arrows), while 2 new, minor peaks eluted at 11.8 min and 14.1 min (Supplementary Fig. 9B, blue arrows). In NAADP-AM digested using PLE, no compound co-eluting with standard NAADP was detected, as shown by magnification of the chromatograms (Supplementary Fig. 9C). Of note, in one single batch from lot #3030795 we observed minute amounts of NAADP, both in digested and undigested NAADP-AM (Supplementary Fig. 10C, upper panel), however, NAADP did not increase upon digestion.

To control enzymatic activity of PLE, commercially available cAMP-AM was digested under same conditions, resulting in almost full conversion to cAMP (Supplementary Fig. 11).

Additionally, KHYG-1 NK cells and Neuro2A neuronal cells were also studied regarding global Ca²⁺ signaling evoked by MASTER-NAADP (Fig. 6). In KHYG-1 NK cells MASTER-NAADP evoked rapid global Ca²⁺ signals, with negligible background signaling observed with MASTER-NADP (Fig. 6A). Quantitative analysis revealed marked difference in Ca²⁺ peak amplitude and percentage of responding cells (Fig. 6B, D), and highly significant statistical differences between active MASTER-NAADP and inactive MASTER-NADP for the number of peaks (Fig. 6C), and the mean responsiveness (Fig. 6D).

In addition to cells of the immune system, T cells or natural killer cells, also the neuronal cell line Neuro2A responded with global Ca²⁺ signaling upon addition of MASTER-NAADP (Fig. 6F–K). Qualitatively, the response to MASTER-NAADP was more similar to the NK cell line KHYG-1 (Fig. 6A left), as compared to Jurkat T cells (Fig. 5A), with rapid and relatively uniform (among individual cells) monophasic and high-amplitude Ca²⁺ signals (Fig. 6F left). Control compound MASTER-NADP did not evoke Ca²⁺ signaling in any of the cells (Fig. 6F right), also more similar to KHYG-1 cells (Fig. 6A right) as compared to Jurkat T cells (Fig. 5A). As for Jurkat T cells, *Hn1l/Jpt2⁻/⁻* Neuro2A cells were created using Crispr/Cas9 technology. Putative null alleles were identified in several Neuro2A cell clones. NGS sequencing revealed frameshift mutations in all alleles in clones 1F3 and 1G4, and accordingly HN1L/

JPT2 protein was not detected in Western blot analysis (Supplementary Fig. 12A). *Hn1l/Jpt2⁻/⁻* Neuro2A clone 1G4 neither responded to active MASTER-NAADP nor to inactive MASTER-NADP (Fig. 6G); likewise, MASTER-NAADP did not stimulate Ca²⁺ signaling in *Hn1l/Jpt2⁻/⁻* Neuro2A clone 1F3 (suppl S12B), confirming also in this cell type that the NAADP mimic benzoic acid C-nucleoside, 2′-phospho-3′F-adenosine-diphosphate activates Ca²⁺ signaling via the NAADP receptor/NAADP binding protein HN1L/JPT2. As for KHYG-1 cells (Fig. 6B), a markedly higher mean value of Ca²⁺ peak amplitude was observed for MASTER-NAADP, as compared to control conditions (Fig. 6H). Quantitative evaluation of number of Ca²⁺ oscillations and mean responsiveness were highly significant when compared between MASTER-NAADP and MASTER-NADP in WT cells (Fig. 6I,K). Likewise, in *Hn1l/Jpt2⁻/⁻* Neuro2A cells these two parameters were extremely low and not significantly different from the values in WT cells stimulated by control compound MASTER-NADP (Fig. 6I,K). Similar results were obtained for the percentage of responding cells (Fig. 6J).

Since stimulation of the different cell types used in this study required different concentrations of MASTER-NAADP to evoke Ca²⁺ signaling, the expression of the NAADP receptor/NAADP binding protein HN1L/JPT2 was compared. Using anti-HN1L/JPT2 antibody HPA041888 (Atlas Antibodies) HN1L/JPT2 was detected as single band at approx. 21 kDa in Jurkat T cells and KHYG-1 cells, while in Neuro2A cells a double band at approx. 24 kDa was observed (Supplementary Fig. 13A); with a different antibody for detection of HN1L/JPT2 (orb1412, Biorbyt) both the double-band at approx. 24 kDa and the band at approx. 20 kDa was observed in Neuro2A cells (Supplementary Fig. 12A). Recombinant HN1l/JPT2 was used as positive control (Supplementary Fig. 13A). When normalized to α-actin no significant differences in expression of HN1l/JPT2 in the three cell types was observed (Supplementary Fig. 13B), indicating that it is not the cellular amount of HN1L/JPT2 that defines the sensitivity of the system. However, post-translational modifications, as seen in Neuro2A cells, might be involved.

## Discussion

Initially, we tried to reproduce the synthesis of NAADP-AM, as described in refs. 28,29. However, this approach led to a complex mixture of compounds and to non-reproducible results, mainly caused by the non-selective reaction site at the NAADP molecule and particularly by the low stability of the AM-ester[59,60]. Moreover, alkylation of the NAADP pyrophosphate moiety by the AM-reagent would lead to high chemical instability of the phosphate anhydride bond due to the neutralization of the negative charge(s). Observations made during synthesis of MASTER-NAADP showed that after the esterification of the carboxylic acid, the N-glycosidic bond in the nicotinic acid moiety became labile. Additionally, the esterification of the 2′-hydroxyl group

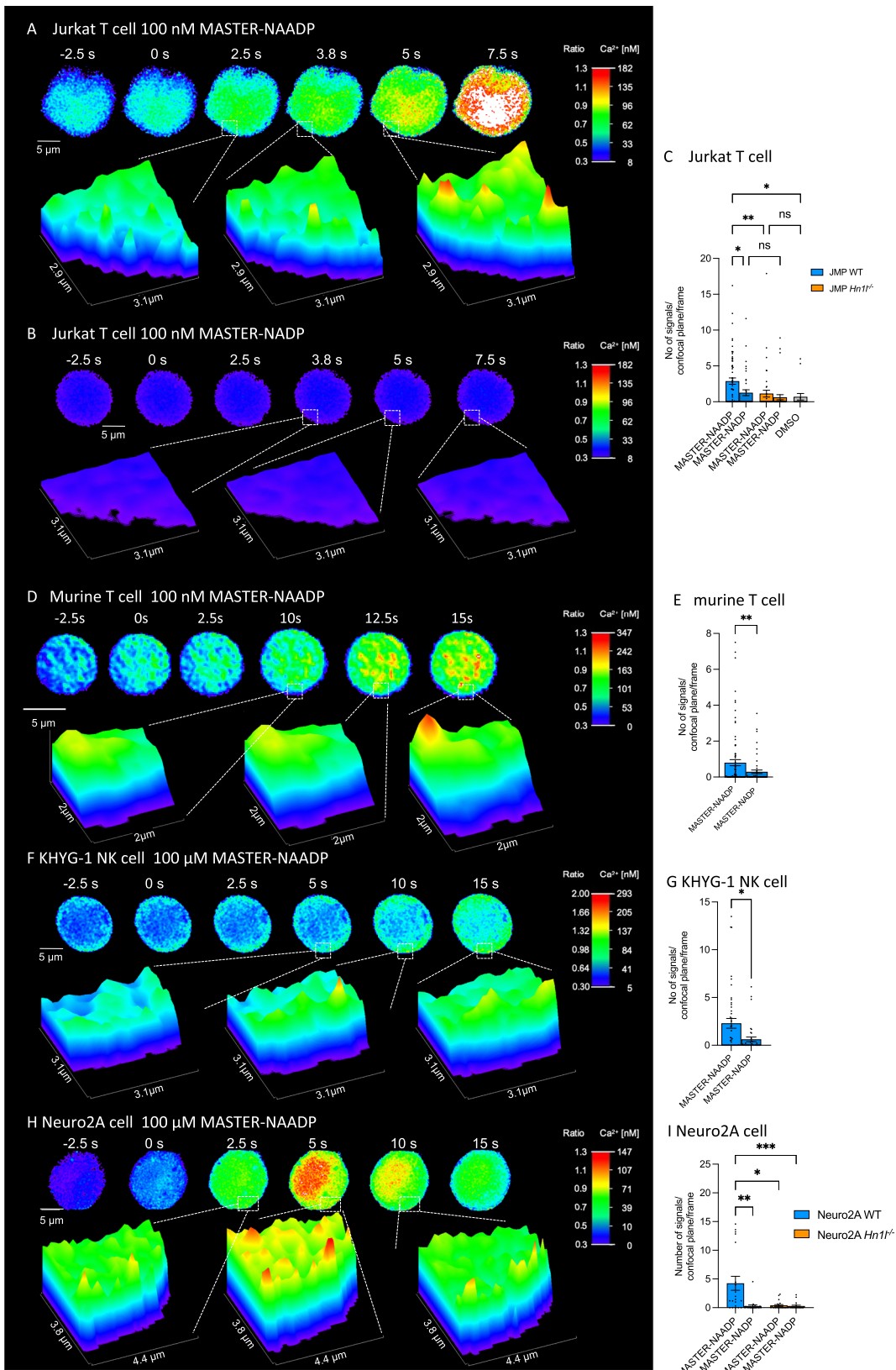

in the adenosine part by a dialkylphosphate led to a high tendency of migration of this esterified phosphate group from the 2′- to the 3′-position or even to cleavage of the ester. All these unwanted properties may also occur after AM-alkylation which finally led to a complex mixture of products. Analysis by HPLC of commercially available NAADP-AM confirmed these problems because all tested lots – some

lots even tested in different batches that were purchased at different time points – showed highly complex mixtures of compounds detected at 260 nm (Supplementary Figs. 9, 10). To our surprise, upon digestion of the commercial NAADP-AM by porcine liver esterases, traces of NAADP were identified only in one of out the four commercially available lots (Supplementary Fig. 10C, 1. batch, right panel);

**Fig. 4 | Ca²⁺ microdomains evoked by MASTER-NAADP in different cell types.**
Representative high-resolution Ca²⁺ images of Jurkat T cells loaded with both Fluo4-AM and Fura-Red-AM after stimulation with 100 nM MASTER-NAADP (**A**) or 100 nM MASTER-NADP (**B**). Number of cells indicated in figure. **A**, **B** (upper panel): Pseudocolor images indicate emission ratios between Fluo-4 and Fura-Red ranging from 0.3 -1.3; ratio data were then converted using external calibration corresponding to 8 to 182 nM [Ca²⁺]ᵢ. Scale bars, 5 µm for whole cells. **A**, **B** (lower panel): Magnified regions as indicated as 3D surface plots. **C** Analysis across the first 15 s of stimulation shown as number of Ca²⁺ microdomains. **C** Data are displayed as mean ± SEM; Jurkat WT MASTER-NAADP, $n = 66$ cells; Jurkat WT MASTER-NADP, $n = 40$ cells; Jurkat *Hn1l/Jpt2*⁻/⁻ MASTER-NAADP, $n = 43$ cells; Jurkat *Hn1l/Jpt2*⁻/⁻ MASTER-NADP, $n = 37$ cells; Jurkat WT DMSO, $n = 17$ cells; nonparametric Kruskal-Wallis test and Dunn's correction for multiple testing *$P < 0.05$; **$P < 0.005$. **D** (upper panel) Representative high-resolution Ca²⁺ images of primary murine T cells loaded with both Fluo4-AM and Fura-Red-AM after stimulation with 100 nM MASTER-NAADP. Pseudocolor images indicate emission ratios between Fluo-4 and Fura-Red ranging from 0.3–1.3; ratio data were then converted using external calibration corresponding to 0 to 347 nM [Ca²⁺]ᵢ. Scale bars, 5 µm for whole cells. **D** (lower panel) Magnified regions as indicated as 3D surface plots. **E** Analysis across the first 15 s of stimulation shown as number of Ca²⁺ microdomains. **E** Data are displayed as mean ± SEM; MASTER-NAADP, $n = 89$ cells; MASTER-NADP, $n = 58$ cells; two-sided, nonparametric Mann-Whitney U test: **$P < 0.005$. **F** (upper panel) Representative high-resolution Ca²⁺ images of human NK cells (KHYG-1) loaded with both Fluo4-AM and Fura-Red-AM after stimulation with 100 µM MASTER-NAADP. Pseudocolor images indicate emission ratios between Fluo-4 and Fura-Red ranging from 0.3–2.0; ratio data were then converted using external calibration corresponding to 5 to 293 nM [Ca²⁺]ᵢ. Scale bars, 5 µm for whole cells. **F** (lower panel) Magnified regions as indicated as 3D surface plots. **G** Analysis across the first 15 s of stimulation shown as number of Ca²⁺ microdomains. **G** Data are displayed as mean ± SEM; MASTER-NAADP, $n = 46$ cells; MASTER-NADP, $n = 39$ cells; two-sided, nonparametric Mann-Whitney U test: *$P < 0.05$. **H** (upper panel) Representative high-resolution Ca²⁺ images of Neuro2A cells loaded with both Fluo4-AM and Fura-Red-AM after stimulation with 100 µM MASTER-NAADP. **H** (lower panel) Magnified regions as indicated as 3D surface plots. Pseudocolor images indicate emission ratios between Fluo-4 and Fura-Red ranging from 0.3–1.3; ratio data were then converted using external calibration corresponding to 0 to 147 nM [Ca²⁺]ᵢ. **I** Analysis across the first 15 s of stimulation shown as number of Ca²⁺ microdomains. **I** Data are mean ± SEM; Neuro2A WT MASTER-NAADP, $n = 20$ cells; Neuro2A WT MASTER-NADP, $n = 23$ cells; Neuro2A *Hn1l/Jpt2*⁻/⁻ MASTER-NAADP, $n = 21$ cells; Neuro2A *Hn1l/Jpt2*⁻/⁻ MASTER-NADP, $n = 20$ cells; nonparametric Kruskal-Wallis test and Dunn's correction for multiple testing *$P < 0.05$; **$P < 0.005$; ***$P < 0.001$. Source data for (**C**, **E**, **G**, **I**) and exact $p$ values are provided as a Source Data file.

moreover, in the same lot #3030795 a very small amount of NAADP was already present before digest (Supplementary Fig. 10C, 1. batch, left panel), indicating spontaneous degradation of NAADP-AM to NAADP, or even remaining starting substrate NAADP within the synthesis product mixture. This finding may be important because, when added to the extracellular solution, NAADP activated Ca²⁺ signaling in intact cells via purinergic P2Y11 receptors was observed[61,62]. Thus, some of the previous results in intact cells stimulated by NAADP-AM[28,30,55–58] may be explained by Ca²⁺ signaling secondary to purinergic P2Y11 receptors.

After the synthesis of the NAADP-AM esters failed in our hands, we decided to develop a total synthesis route towards the membrane-permeable NAADP precursors which are modified with our previously developed acyloxybenzyl moieties that are cleaved in the cytosol by esterases/lipases[31,63,64]. Retrosynthetically a convergent approach was used in which the two nucleotidic parts were synthesized first and coupled at the end to form the pyrophosphate bridge. First, attempts were made to use nicotinic acid which was esterified at the carboxylic group. However, this led to a marked increase in lability of the glycosidic bond, which was incompatible with the later reaction conditions. Therefore, we decided to replace the *N*-glycosidic bond by a C-nucleosidic bond ("northern" part). Also, in the adenosine ("southern" part) the presence of the 3'-OH group and the 2'-phosphate ester group resulted in instabilities. Over time, degradation of the phosphate ester was detected by HPLC analysis due to 2',3'-phosphate group migration. Since the 2'-phosphate is essential for NAADP's Ca²⁺ mobilizing activity[50], we decided to replace the hydroxyl group at the 3'-position by the bioisosteric fluorine atom. This replacement may change the conformational properties of the molecule to some degree, but completely avoided the migration of the phosphate group in aqueous incubation studies. While the 3'-OH-comprising compound showed a marked conversion, the 3'-fluorinated derivative proved to be completely stable. In case of the control compound, MASTER-NADP, no laborious 3'-fluorination was included; this resulted in production of a mixture of the 2'-phosphate and 3'-phosphate isomers of deMASTER-NADP upon digestion with porcine liver esterases, as described[48,49]. Since deMASTER-NADP does not contain the 3-COO⁻ group at the C-nucleoside and thus does not possess Ca²⁺ releasing activity, the 2',3'-phosphate migration can be tolerated in this control compound.

Structural activity data for NAADP were initially obtained mainly in the sea urchin egg homogenate system. Already, in 1997 Lee and Aarhus demonstrated the significance of three parts of NAADP: the 3-carboxyl group of the nicotinic acid moiety, the 2'-phosphate of the adenosine moiety, and the 6-amino/imino group of the nucleobase adenine[50]. In particular, they showed that either moving the 3-carboxyl group to the 4-position of nicotinic acid or replacement by an uncharged group (3-carbinol-derivative) completely abolished Ca²⁺ release[50]. Replacement of the 3-carboxyl group by a 3-sulfonic acid group resulted in a weaker agonist[50]. The significance of the 2'-phosphate was demonstrated by replacement using either 2',3'-cyclic-phosphate or 3'-phosphate at the adenosine ribose; both modifications shifted the EC₅₀ towards higher agonist concentrations, but did not completely abolish Ca²⁺ release activity[50]. The role of the 6-amino/imino group of the purine base adenine was assessed by using the hypoxanthine derivative of NADP, NHDP, as substrate to produce nicotinic acid hypoxanthine dinucleotide phosphate (NAHDP) via the base-exchange mechanism. NAHDP released Ca²⁺ from sea urchin egg homogenate only at concentrations ≥5 µM, while the EC₅₀ for NAADP was approx. 10 nM[65]. When the 4-position of the nicotinic acid moiety in NAADP was substituted, EC₅₀ was shifted to higher concentrations; in contrast, different substitutions at the 5-position showed good agonist activity[65]. Of note, both 4- and 5-modified caged NAADP derivatives showed Ca²⁺ mobilizing activity in mammalian SKBR cells upon microinjection and flash photolysis, suggesting differences in agonist recognition by NAADP receptors of sea urchin eggs vs higher eukaryotic cells[66]. NAADP derivatives modified in the 8-position of adenine, 8-ethynyl-NADP, 8-ethynyl-NAADP, and 5-N₃-8-ethynyl-NAADP did not significantly decrease agonist activity in sea urchin egg homogenates[67].

deMASTER-NAADP contains all relevant moieties for Ca²⁺ mobilizing activity: the 3-carboxyl group of the nicotinic acid moiety, the 2'-phosphate of the adenosine moiety, and the 6-amino/imino group of adenine. However, as discussed above, specific structural variations had to be introduced: (i) the benzoic acid C-nucleoside to overcome the lability of the *N*-glycosidic bond of nicotinic acid during synthesis, and (ii) the 3'F-adenosine to inhibit migration of the 2'-phosphate during synthesis or de-esterification. Further, modeling studies (Fig. 2C) indicate a highly similar overall structure of NAADP and deMASTER-NAADP in solution. As result, deMASTER-NAADP released Ca²⁺ in permeabilized Jurkat T cells and increased the open probability of RYR1 in lipid planar bilayer recordings (Fig. 3). Of note, both experiments required presence of NAADP receptor HN1L/JPT2. The lipophilic precursor MASTER-NAADP evoked both local and global Ca²⁺ signaling in intact cells whereas deMASTER-NADP did not (Figs. 4, 5, 6).

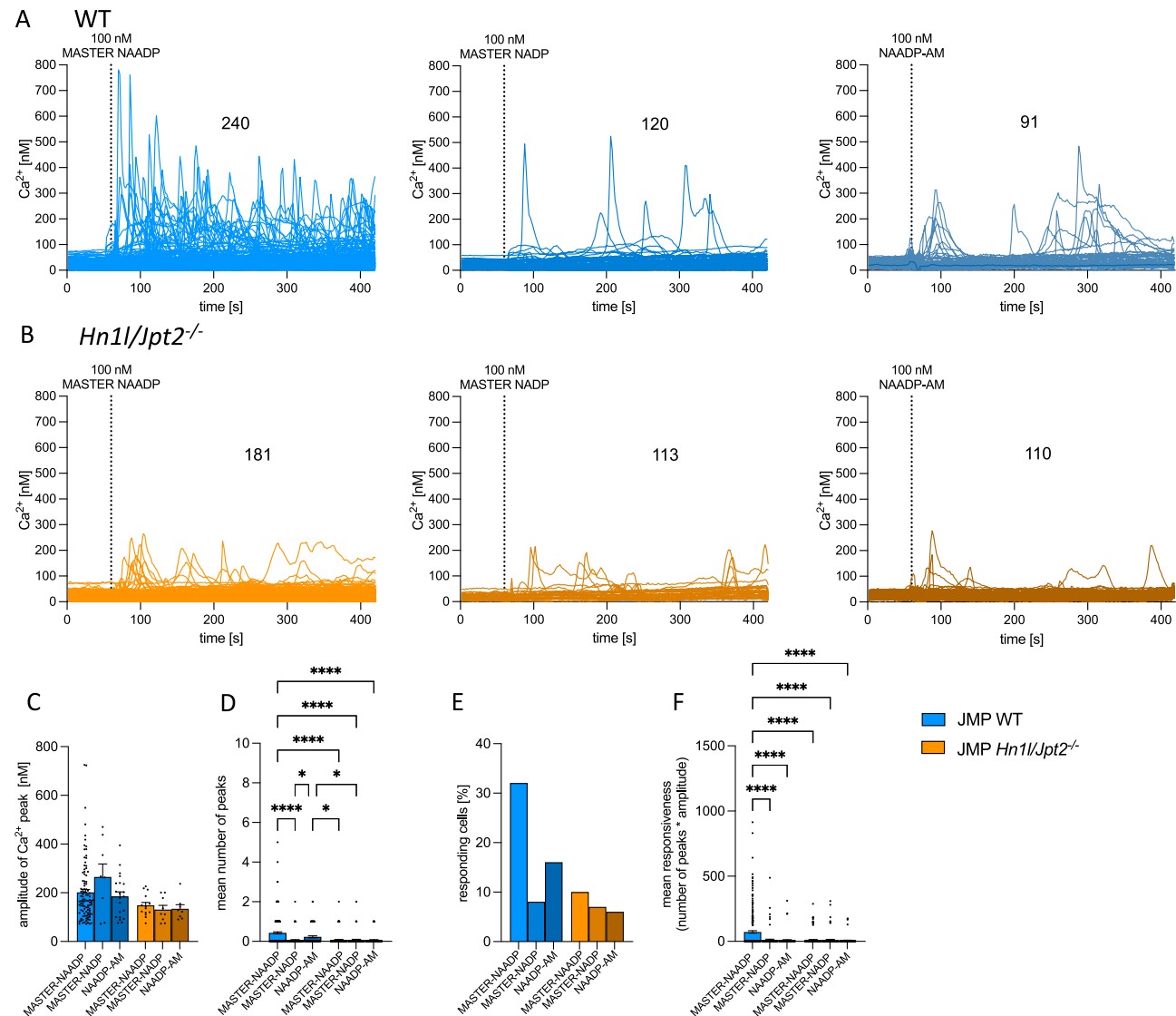

**Fig. 5 | Global Ca²⁺ responses evoked by MASTER-NAADP in Jurkat T cells.** Global Ca²⁺ signaling was analyzed in Jurkat WT T cells (**A**) and Jurkat *Hn1l/Jpt2⁻ᐟ⁻* T cells (**B**), loaded with Fura2-AM. MASTER-NAADP (100 nM), MASTER-NADP (100 nM), or NAADP-AM (100 nM) were added as indicated. Experiments were carried out at 37 °C and SOCE was blocked by pre-incubation of 50 µM Synta66 for 5 min prior to imaging. Number of cells indicated in figure. Aggregated data are mean amplitude per peak (**C**), mean number of Ca²⁺ peaks (**D**), percentage of responding cells (**E**), and calculation of the mean responsiveness (number of peaks * amplitude; **F** presented as mean ± SEM; Jurkat WT MASTER-NAADP, $n = 240$ cells; Jurkat WT MASTER-NADP, $n = 120$ cells; Jurkat WT NAADP-AM, $n = 91$ cells; Jurkat *Hn1l/Jpt2⁻ᐟ⁻* MASTER-NAADP, $n = 181$ cells; Jurkat *Hn1l/Jpt2⁻ᐟ⁻* MASTER-NADP, $n = 113$ cells; Jurkat *Hn1l/Jpt2⁻ᐟ⁻* NAADP-AM, $n = 110$ cells. Nonparametric Kruskal-Wallis test and Dunn's correction for multiple testing *$P < 0.05$; ****$P < 0.0001$. Source data and exact $p$ values are provided as a Source Data file.

The major goal of this study was to produce a membrane-permeable, stabilized, biocleavable protected precursor of NAADP (MASTER-NAADP). MASTER-NAADP was used to stimulate different cell types, both cell lines and also primary cells. In general, we observed better results at 37 °C, as compared to room temperature, probably allowing for more rapid release of active NAADP mimic benzoic acid C-nucleoside, 2'phospho-3'F-adenosine-diphosphate. The extracellular concentration required to evoke Ca²⁺ signaling ranged from 100 nM in T cells to 100 µM in KHYG-1 NK cells or Neuro2A neuronal cell line. This variability is likely explained by (i) the diffusion rate of the MASTER-compounds through the plasma membrane, (ii) different levels of endogenous esterase activity, (iii) metabolic stability of deMASTER-NAADP, and (iv) expression of xenobiotic efflux pumps. Differences in protein expression levels of HN1L/JPT2 in the different cell lines were ruled out as potential reasons (Supplementary Fig. 13). After optimization of temperature and concentration, we observed

local Ca²⁺ signals comparable to the TCR/CD3 and CD28 evoked Ca²⁺ microdomains observed upon directed stimulation of T cells using beads coated with antibodies against CD3 and CD28[11,17,20,21]. KHYG-1 NK cells also responded with Ca²⁺ signaling to acute stimulation with MASTER-NAADP, confirming NAADP-dependent Ca²⁺ signaling in IL-8 stimulated lymphokine-activated killer cells[8,13]. MASTER-NAADP was also successfully applied to non-hematopoietic cells, such as the Neuro2A neuronal cell line. MASTER-NAADP addition evoked local and global Ca²⁺ signals in Neuro2A cells, while MASTER-NADP was completely without effect for both signal types, confirming that neurons utilize NAADP for Ca²⁺ signaling, as previously shown for control of social behavior[68].

Of note, in both Jurkat T cells and Neuro2A cells, clones lacking expression of the *Hn1l/Jpt2* gene encoding HN1L/JPT2, the recently discovered NAADP receptor/NAADP binding protein[17,18], did not respond to MASTER-NAADP, confirming that the endogenous

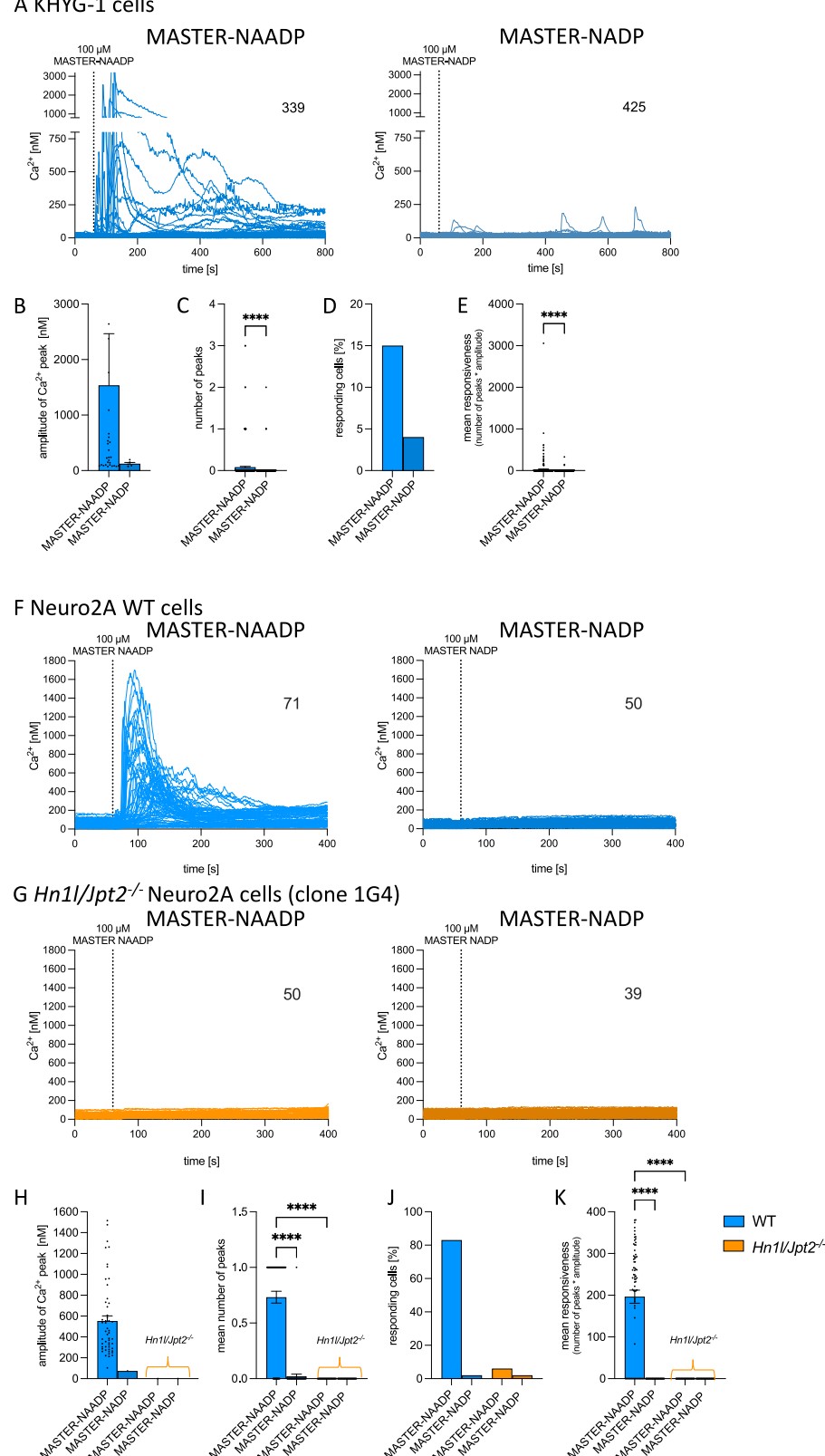

signaling pathway of NAADP is mimicked by its derivative deMASTER-NAADP.

A crucial point for us was whether MASTER-NAADP can be reliably used to study NAADP mediated $Ca^{2+}$ signaling processes in different cell types as a bona fide membrane permeable, stabilized, biocleavably protected precursor. In fact, MASTER-NAADP's chemical stability,

release of the NAADP mimetic benzoic acid C-nucleoside, 2'phospho-3'F-adenosine-diphosphate, and signaling via NAADP receptor/NAADP binding protein HN1L/JPT2 evoked $Ca^{2+}$ mobilization in different cell types. Thus, MASTER-NAADP appears generally applicable to investigate NAADP signaling and its downstream events in cells. As a limitation, we observed that quantitatively there are differences regarding

**Fig. 6 | Global Ca²⁺ responses evoked by MASTER-NAADP in KHYG-1 NK cells and Neuro2A cells.** Global Ca²⁺ signaling was analyzed in Fura2-loaded KHYG-1 cells (**A–E**), Neuro2A WT or Neuro2A *Hn1l/Jpt2⁻/⁻* cells (**F–K**). **A** MASTER-NAADP (100 µM) or MASTER-NADP (100 µM) were added at RT as indicated. Number of cells indicated in figure. Aggregated data are mean amplitude per peak (**B**), mean number of Ca²⁺ peaks (**C**), responding cells (**D**), and calculation of the mean responsiveness (number of peaks * amplitude; **E**; as mean ± SEM; MASTER-NAADP, n = 339 cells; MASTER-NADP, n = 425 cells; two-sided, nonparametric Mann-Whitney U test: ****P < 0.0001. **F** and **G** MASTER-NAADP (100 µM) or MASTER-NADP (100 µM) were added at RT as indicated to Neuro2A WT (**F**) or Neuro2A *Hn1l/Jpt2⁻/⁻* cells (**G**). Number of cells is indicated in figure. Aggregated data are mean amplitude per peak (**H**), mean number of Ca²⁺ peaks (**I**), responding cells (**J**), and calculation of the mean responsiveness (number of peaks * amplitude; **K**; as means ± SEM; Neuro2A WT MASTER-NAADP, *n* = 71 cells; Neuro2A WT MASTER-NADP, *n* = 50 cells; Neuro2A *Hn1l/Jpt2⁻/⁻* clone 1G4 MASTER-NAADP, *n* = 50 cells; Neuro2A *Hn1l/Jpt2⁻/⁻* clone 1G4 MASTER-NADP, *n* = 39 cells; Nonparametric Kruskal-Wallis test and Dunn's correction for multiple testing ****P < 0.0001. Source data are and exact *p* values provided as a Source Data file.

the magnitude of Ca²⁺ mobilization by MASTER-NAADP vs MASTER-NADP between the cell types studied. However, the fact that MASTER-NADP is available as relevant control allows a more precise investigation of cellular NAADP signaling in the future.

## Methods

### Ethics
All mice experiments were approved by the Animal Welfare Officers of the University Medical Center Hamburg-Eppendorf (UKE) and Behörde für Gesundheit und Verbraucherschutz Hamburg (ORG934).

### Chemistry
Most reactions were performed in oven-dried glassware, under nitrogen atmosphere, and with dry solvents. THF, CH₂Cl₂, acetonitrile and pyridine were dried with a MBraun (MB SPS-800) Solvent System and stored over nitrogen and molecular sieves (3 or 4 Å). Further dry solvents which are not listed were purchased from Acros Organics and dried over molecular sieves. All starting materials and reagents were purchased from Sigma Aldrich, TCI, Acros, ABCR or Carbosynth and were used without further purification. *Ultrapure Water* was obtained from a Sartorius arium pro system (Sartopore 0.2 µm, UV) and HPLC-grade CH₃CN, used for automated RP-18 chromatography and HPLC analysis, was purchased from VWR or Honeywell. *Column chromatography* was carried out with technical grade solvents, which were distilled before use, and silica gel (0.04–0.063 mm) from Macherey-Nagel. Precoated thin-layer-sheets (ALU- GRAM® Xtra SIL G/UV 254, Macherey-Nagel) were used for *thin-layer-chromatography*. The UV-active compounds were detected using a UV lamp at a wavelength of 254 nm. The non-UV-active compounds could be visualized using various staining reagents (cerium ammonium nitrate, potassium permanganate and vanillin) and followed by heat treatment. *Automated RP-18 flash chromatography* was performed using prepacked MN RS 40 C18ec columns on an Interchim Puriflash 430. For all automated *RP-18 chromatography*, a gradient from 100 % ultrapure water to 100 % CH₃CN within 20 min was used (flowrate: 20 mL/min). All NMR solvents were purchased from Euriso-Top or Deutero. *NMR spectra* were recorded on Bruker instruments Fourier HD 300, Avance I 400, DRX 500, or Avance III 600 at room temperature. All proton and carbon NMR spectra were calibrated by the respective solvent signal. *High-resolution mass spectra* were measured with an Agilent 6224 ESI-TOF instrument, detailed information can be found in the section mass spectrometry. IR spectra were obtained on a Bruker Alpha IR spectrometer.

Details regarding synthesis and characterization of chemical intermediates and final products are found in the Supplementary Methods section of the Supplementary Information File.

### General synthetic procedure A: synthesis of 4-(hydroxymethyl) phenylalkanoates and 4-(hydro-xymethyl) phenylalkylcarbonates
Under nitrogen atmosphere, 4-hydroxybenzyl alcohol (1.1 eq. - 1.2 eq.), dry Et₃N (1.0 eq. - 1.2 eq.), and a catalytic amount of 4-dimethylaminopyridine (DMAP) were dissolved in dry CH₂Cl₂ or dry THF and cooled to 0 °C. A solution of acid chloride or alkyl chloroformate (1.0 eq.) in dry CH₂Cl₂ or dry THF was added dropwise, and the mixture was stirred for 2–3 h at 0 °C or room temperature. The crude product was filtered and the filtrate was concentrated in vacuo. The residue was diluted with CH₂Cl₂ and washed once with saturated sodium hydrogen carbonate solution and once with H₂O. The organic layer was dried with Na₂SO₄ and concentrated in vacuo. The crude product was purified by column chromatography.

### General synthetic procedure B: synthesis of phosphoramidites
In a nitrogen atmosphere, dichloro-*N,N*-di*iso*propylaminophosphoramidite (1.0 eq.) was dissolved in dry THF and cooled to −20 °C – 0 °C. A solution of the respective alcohol (2.0 eq. – 2.2 eq.) and dry Et₃N (2.2 eq.- 2.3 eq.) in dry THF were added dropwise. The reaction mixture was stirred for 16 h at room temperature. The suspension was filtered, and the filtrate was concentrated under reduced pressure. The crude product was purified by column chromatography.

### General synthetic procedure C: Synthesis of 5'-phosphates
The nucleoside (1.0 eq.) was dissolved in dry trimethyl phosphate under a nitrogen atmosphere and the solution was cooled to −5 °C. The solution was mixed with tributylamine, 1,8-bis(*N, N*-dimethylamino) naphthalene, or activated mole sieve (3 Å) (2.5 eq.). After the solution was cooled to 0 °C, P(O)Cl₃ (2.0–4.0 eq.) was added slowly. The reaction solution was then stirred for 3–16 h at 0 °C. To stop the reaction, a 1 M TEAB buffer solution was added and stirred for 15 min at room temperature. The solution was extracted three times with hexane. The aqueous layer was purified using automatic RP-18 column chromatography (H₂O/CH₃CN gradient) and finally lyophilized.

### General synthetic procedure D: Coupling to the NAADP analog
The nicotinic acid unit (1.0 eq.) or its analogs was dissolved in dry acetonitrile and cooled to 0 °C. In a separate flask, dry Et₃N (16.0 eq.) was added to dry CH₃CN and then TFAA (10.0 eq.) was slowly added. After stirring at 0 °C for 10 min, the solution was dropped slowly to the solution of nicotinic acid analogs. The solution was stirred at room temperature for 15 min and the solvent was removed in vacuo. The residue was then dissolved in dry CH₃CN, cooled to 0 °C, and 1-methylimidazole (NMI) (6.0 eq.) was added. The reaction solution was stirred for 15 min at room temperature. Then, the corresponding adenosine derivative (0.9 eq.) was coevaporated three times with dry CH₃CN and then added dissolved in dry CH₃CN. The mixture was stirred for 3 h at room temperature. To terminate the reaction, a 1 M TEAB buffer solution was added and stirred for 15 min at room temperature. The aqueous layer was purified by RP-18 column chromatography (H₂O/CH₃CN gradient) and finally lyophilized.

### Reagents for biochemistry and cell biology
Fura2-AM, Fluo4-AM, and Fura-Red-AM were obtained from Life Technologies. All dyes were dissolved in dimethyl sulfoxide (DMSO), divided into aliquots, and stored at −20 °C until required for use. Synta66 was purchased from aobius inc., dissolved and aliquoted in 50 mM stocks in DMSO and stored at −20 °C.

NAADP-AM was obtained from AAT Bioquest and stored lyophilized at −80 °C until use. MASTER-NAADP and MASTER-NADP

lyophilised aliquots were dissolved in DMSO (stock concentration 10 mM). NAADP-AM, MASTER-NAADP and MASTER-NADP were aliquoted into 'single-use aliquots' and stored at −80 °C until use.

## Cell culture

Wild type (WT) and *Hn1l/Jpt2−/−* clone 2 Jurkat T cells were cultured in RPMI (1640) medium (Gibco, Life Technologies) containing 25 mM HEPES and GlutaMAX-1, which was further supplemented with 100 U/mL penicillin, 100 µg/mL streptomycin and 7.5% (v/v) newborn calf serum (NCS) (Biochrom, Merck Millipore). For culturing NK cell line KHYG-1, RPMI (1640) medium (Gibco, Life Technologies) containing 25 mM HEPES and GlutaMAX-1 further supplemented with 10% (v/v) fetal calf serum (FCS) and 10 ng/mL IL-2 (PeproTech) was used. Neuro2a cells were maintained in DMEM medium including GlutaMAX-1 and additionally containing 100 U/mL penicillin, 100 µg/mL streptomycin and 10 % (v/v) fetal calf serum (FCS) (Biochrom, Merck Millipore). All cells were kept at 37 °C and 5 % CO$_2$ in the incubator. For Jurkat T cells, as well as KHYG-1 cells, media was changed every second to third day. Cell density was maintained between 0.1 to 0.8 cells/mL for KHYG-1 cells, 0.3 and 1.3 million cells/mL for Jurkat T cells and Neuro2a cells. Neuro2a cells were split every three days.

## Animals

Spleen and lymph nodes of WT mice (C57BL/6 J; *Mus musculus*) were used for studies with primary murine CD4$^+$ T cells. Housing was performed under standardized conditions at a 12/12 h light-dark cycle with food and water supply *ad libitum* in the animal facility of the University Medical Center Hamburg-Eppendorf (UKE). All mice experiments were approved by the Animal Welfare Officers of UKE and Behörde für Gesundheit und Verbraucherschutz Hamburg (ORG934).

## Isolation of primary murine T cells

Spleens and lymph nodes were immediately used for isolation of primary murine CD4$^+$ T cells after dissecting them from the mice. RPMI, 7.5 % (v/v) newborn calf serum (NCS), 1% (v/v) penicillin/streptomycin was used as isolation medium. The tissue was disrupted using a syringe plunger and cell strainer (40 µm nylon). After transfer to a 50 mL tube, the suspension was centrifuged at 4 °C (300 g, for 5 min) and the supernatant was then discarded. 5 mL of cold ACK buffer (4.3 g ammonium chloride, 0.5 g KHCO$_3$, 0.0186 g Na$_2$-EDTA in 400 mL H$_2$O, pH 7.2 to 7.4) was used for the lysis of erythrocytes, which was stopped by the addition of medium. After centrifugation (4 °C, 300 g, for 5 min), the supernatant was discarded, followed by resuspension of the pellet in 2 mL PBS. Further isolation of CD4$^+$ T cells was carried out using a negative selection kit (EasySep™ Mouse T Cell Enrichment Kit, STEMCELL Technologies, Germany) following instructions of the manufacturer. Cell purity was typically 95% CD4 + T cells and was determined by immunostaining with fluorescein isothiocyanate–conjugated anti-mouse

TCRβ antibody (clone H57-597, BioLegend) and measured with a FACSCalibur flow cytometer (BD Biosciences).

## Global ratiometric Ca$^{2+}$ imaging using Fura2-AM

For ratiometric Ca$^{2+}$ imaging, all cells except Neuro2a cells were loaded for 40 min at 37 °C with the fluorescent Ca$^{2+}$ indicator Fura2-AM (4 µM) diluted in RPMI medium. Neuro2a cells were incubated at RT for 20 min with 2 µM Fura2-AM in DMEM medium additionally containing 0.1% Pluronic F-127 (Sigma-Aldrich Chemie GmbH, Steinheim, Germany). To all cell types 2 ml of fresh medium was added after 15 min of incubation. Cells were then washed with Ca$^{2+}$ buffer (140 mM NaCl, 5 mM KCl, 1 mM MgSO$_4$, 1 mM CaCl$_2$, 1 mM NaH$_2$PO$_4$, 20 mM HEPES, 5.5 mM glucose, pH 7.4, sterile filtered) before being re-suspended in Ca$^{2+}$ buffer. For adhesion during live-cell imaging of Jurkat T cells, primary murine CD4$^+$ T cells and KHYG-1 cells, coverslips were coated with bovine serum albumin (BSA) (5 mg/mL, Merck), followed by

coating with poly-L-lysine (PLL) (0.1 mg/mL, Merck). Rubber O-rings were sealed with silicone grease (Kurt Obermeier GmbH & Co. KG, Germany) and glued to the coverslips to create a chamber for seeding the cells. Approximately 100.000 cells per slide were seeded a few minutes prior to imaging. In contrast, uncoated 8 well µ-slides (IBIDI GmbH, Germany) were used to place 10.000 Neuro2a cells per well 72 h before imaging.

Measurements were either performed at RT or in an incubation chamber at 37 °C. Additionally, Jurkat T cells were incubated with 50 µM Synta66 (Merck) for 5 min prior to imaging to block SOCE through inhibition of Orai1. Next, slides were mounted on a Leica IRBE microscope (40-fold magnification). Images were taken with an electron-multiplying charge-coupled device camera (EM-CCD; C9100-12, Hamamatsu) and a Sutter DG-4 was used as a light source (Filter-set (nm) excitation (ex): HC340/26, HC387/11; beam splitter (bs): 400DCLP; em: 510/84) at 0.5 ratios/s. Image acquisition was performed with Volocity software (version 6.6.2; PerkinElmer Inc.). Stimulation of cells with either commercial NAADP-AM, MASTER-NAADP, MASTER-NADP as a control compound or DMSO as a vehicle control was performed 60 s after starting the acquisition. Depending on the cell type and experimental procedure, the final concentrations of NAADP-AM, MASTER-NAADP or MASTER-NADP were either 100 nM, 10 µM or 100 µM. As a positive control either thapsigargin (1.67 µM) or OKT3 (1 µg/mL) was added after 7 min. For data processing Fiji (ImageJ Version 2.1.0/1.53c) was used. First, the ratiometric raw file was split into respectively 340 nm and 380 nm channels. Next, background correction was performed and cells were selected by drawing a round-shaped ROI (region of interest) and for both channels' fluorescence intensity was measured over time. The resulting values for each cell in the 340 nm and 380 nm channels were used to calculate the ratio 340 nm/380 nm in Excel (Microsoft, USA). To obtain Ca$^{2+}$ concentration, calibration was performed using a high concentration of 5 µM Ionomycin (Santa Cruz Biotechnology, USA), to measure the maximal ratio value (Rmax). Furthermore, the minimal ratio value (Rmin) was determined by chelating free Ca$^{2+}$ using 4 mM EGTA (Sigma-Aldrich, Germany) before incubation with Ionomycin. Calculation of Ca$^{2+}$ concentration was performed using the equation in ref. [69]. Furthermore, using Prism the mean Ca$^{2+}$ concentration per peak, the number of peaks as well as the percentage of responding cells, defined as an increase of at least 120 nM above baseline and lasting for at least 2 time points, was calculated. Additionally, the mean responsiveness refers to the mean Ca$^{2+}$ concentration multiplied by the number of peaks of the respective cell.

## HPLC purity analysis of NAADP-AM, MASTER-NAADP, or MASTER-NADP and deprotection by esterase treatment

Commercially available NAADP-AM (AAT Bioquest), cAMP-AM (Biolog), MASTER-NAADP or MASTER-NADP were incubated with 5 U/mL of porcine liver esterase (PLE) (Merck) for 5 min at 37 °C. Reversed-phase (RP)-HPLC was used to analyze the purity of educts and to separate the reaction products by using a 250 mm × 4.6 mm C8 Luna column (5 µm particle size, Phenomenex) as stationary phase. The mobile phase consisted of HPLC buffer A (20 mM KH$_2$PO$_4$, pH 6.0) and B (10% buffer A, 90% methanol), and elution of nucleotides from the column occurred with increasing methanol content in the mobile phase (0.0 min [100.0 % buffer A], 5.0 min [100.0 % buffer A], 27 min [100 % buffer B], 32.0 min [100% buffer B], 34.0 min [100.0 % buffer A], 45.0 min [100.0 % buffer A]). The flow rate was set to 0.8 mL/min. A diode-array detector (DAD, Agilent Technologies) was used to detect nucleotides at 260 nm. For each experiment different samples were analysed: (i) 250 pmol NAADP- or 1 nmol cAMP-standard (Biolog), (ii) 1 nmol cAMP-AM, 5 nmol NAADP-AM, 5 nmol MASTER-NAADP or 10 nmol MASTER-NADP, (iii) 1 nmol cAMP-AM, 5 nmol NAADP-AM, 5 nmol MASTER-NAADP or 10 nmol MASTER-NADP after PLE digest, and (iv) PLE alone.

Furthermore, after digestion of MASTER-NAADP and MASTER-NADP with PLE, product peaks were collected. Therefore, a 250 mm × 4.6 mm C8 Luna column (5 µm particle size, Phenomenex) was used as stationary phase. The mobile phase consisted of different compositions of HPLC buffer A (97% 50 mM triethylamine acetate buffer, 0.3% (v/v) acetic acid, pH 5.5 and 3% methanol) and B (3% buffer A, 97% methanol). The gradient was as follows: 0.0 min [100.0 % buffer A], 7.0 min [100.0% buffer A], 57 min [100% buffer B], 62.0 min [100% buffer B], 70.0 min [100.0% buffer A], 75.0 min [100.0 % buffer A]. The flow rate was set to 0.8 mL/min. The major peak after digestion of MASTER-NAADP by PLE was collected and subjected to mass spectrometry (see below). Data were processed by the OpenLAB CDS ChemStation Edition C.01.05 data acquisition software from Agilent.

## HPLC of intermediates and end products during chemical synthesis

HPLC Chromatograms were recorded on an Agilent 1260 Infinity II with the following components: $C_{18}$ column (Macherey-Nagel, EC 125/3 Nucleodur 100-5), G7129A Vialsampler, G7111A Quat Pump, G7116A MCT, G7117C DAD HS. UV detection was performed at wavelengths of 240 nm, 250 nm, 260 nm and 270 nm with the following method: 0–25 min 5 % $CH_3CN$ in 2 mL aq. tetrabutylammonium acetate buffer (pH 6.0) 5 % to 100 % $CH_3CN$, 1 mL/min.

## High-resolution mass spectrometry (HRMS and electrospray ionization)

The samples were dissolved in water (LC-MS grade) or organic solvents and measured with electron-spray ionization-time of flight mass spectrometry (Agilent 6224 ESI-TOF instrument). The measurements were done in positive and negative mode with a mass range of m/z 110-3200 and a rate of 1.03 spectra/s. The gas temperature was set to 325 °C and the drying gas flow to 10 L/min. Date evaluation was performed using MestreNova (version 14.0.1-235559) software.

## Molecular modeling

Molecular Modeling of NAADP and MASTER-NAADP was performed with SCHRÖDINGERS MAESTRO (Version 12.6). Ligands were generated using the LigPrep module of MAESTRO. The overlay of the structures was performed with the ligand alignment module of MAESTRO.

## $Ca^{2+}$ microdomain imaging

Detection of $Ca^{2+}$ microdomains after addition of MASTER-NAADP or MASTER-NADP, as well as image processing and analysis of these data was performed as described in detail in ref. 70.

The loading protocol was similar for all cell types. Shortly, cells were loaded with Fluo4-AM (10 µM) and Fura-Red-AM (20 µM) (Life Technologies) for 50 min at RT. Next, cells were washed and taken up in $Ca^{2+}$ buffer. The procedure for coating the slides and seeding the cells was performed similarly to that described for the global $Ca^{2+}$ imaging procedure was carried out on a wide-field microscope using a Dual-View module (Optical Insights, PerkinElmer Inc.) to split the emission wavelengths (filters: excitation (ex), 480/40; beam splitter (bs), 495; emission 1 (em1) 1, 542/50; em2, 650/57). For image acquisition a 100x magnification and an exposure time of 25 ms was used. During the first min the frame rate was set to 1 frame per 5 s and the remaining 2 min were imaged at maximum frame rate (40 frames/s). After 60 s the compound was added to evoke $Ca^{2+}$ microdomain formation. Postprocessing and detection of $Ca^{2+}$ microdomains were performed as described in ref. 70 using MATLAB (version 2022a). The first step during the postprocessing was a frame-by-frame additive bleaching correction of FuraRed using a biexponential decay curve. To get digital confocal images, a Lucy-Richardson deconvolution with a maximum number of 4 iterations, independent of each frame, was applied to both channels using an analytically generated point spread

function (PSF). The next step was to spatially align both channels, remove the image background by semiautomated thresholding and compute a ratio image sequence by frame-by-frame pixel-wise division of Fluo4 and FuraRed intensities. Calibration is based on Grynkiewicz et al.[69] with a $K_d$ of Fluo4 and FuraRed of $408 \pm 12$ nM[20]. Since an additive frame-by-frame bleaching correction is applied for Fura-Red, the mean Fura-Red fluorescence remains stable throughout each measurement and is included in the calibration as Fura-Red$_{t0}$ (initial mean Fura-Red fluorescence). Subcellular $Ca^{2+}$ microdomains are defined as small, compact connected sets of pixels with high $[Ca^{2+}]_i$ values. To discriminate between signal and noise, noise was determined in a cell-free system in EGTA buffer ($[Ca^{2+}] = 100$ nM) to calculate a threshold for the $Ca^{2+}$ microdomain detection. The threshold for Jurkat cells is 72 nM and for primary T cells, N2A cells and KHYG-1 cells is 112.5 nM. Next, on a frame-by-frame basis the frame-specific mean $[Ca^{2+}]_i$ value was calculated and a $Ca^{2+}$ microdomain is counted as one if the $Ca^{2+}$ microdomain consists of at least 6 and maximally 20 pixels, that is limited by the spatial resolution of 368 nm, and if the $[Ca^{2+}]_i$ value of the $Ca^{2+}$ microdomain is ≥ the frame specific mean plus the respective threshold. The starting time of cell activation after the addition of MASTER-NAADP was determined automatically and defined as the beginning of a linear ascent of the logarithmized mean cell signal (i.e., the beginning of an exponential signal increase). Furthermore, for the localization of MASTER-NAADP evoked $Ca^{2+}$ microdomains Jurkat and KHYG-1 cells were shape-normalized and dartboard projections were generated to visualize $Ca^{2+}$ microdomain mean data, as described in ref. 51. To analyze pure MASTER-NAADP $Ca^{2+}$ microdomains, responses to MASTER-NADP (control compound) were subtracted from MASTER-NAADP signals. Data analysis was performed with Excel (Microsoft, USA) and Prism 10 (GraphPad Software, USA).

## Recombinant production of HN1L/JPT2

HN1L/JPT2 protein was produced as described recently[17] using the extracted plasmids pETM33 with His-tag and GST-tag. The *hn1l/jpt2* gene (accession number, Q9H910) was codon-optimized for bacterial expression and cloned in XL1-blue bacteria (New England Biolabs). Rosetta2 E. coli strain was transformed with plasmid DNA. The culture after isopropyl β-D-1-thiogalactopyranoside induction was further incubated at 24 °C for 24 h at 180 rpm. Bacteria were lysed by ultrasonication on ice (3 times for 3 min, 70% power, 50% cycle) using Sonopuls GM 70 (Bandelin Electronic). Recombinant HN1L/JPT2 was then purified from the supernatant using the ÄKTA pure™ 25 system (Cytiva) with HisTrapTM FF 5 mL column (Cytiva). The His-tag was cleaved using PreScission Protease (provided by Dr. Susanne Witt, CSSB, Hamburg, Germany) and removed with the previously mentioned system and column. Size exclusion chromatography was performed with the same system and with a HiLoad SuperdexTM 75 pg (Cytiva) column. The protein was concentrated using an Amicon® Ultra-15 Centrifugal Filter 10 kDa MWCO (Merck). Concentrations of the purified protein were determined by measuring absorbance at 280 nm using NanoDrop Onec (Thermo Fisher Scientific). To check the purity of recombinant HN1L/JPT2 SDS-PAGE was performed and the identity was assessed using anti-HN1L antibodies (1:5000, ab200587 abcam, and 1:5000, HPA041888, Atlas Antibodies) on a Western blot.

## $Ca^{2+}$ release in permeabilized Jurkat T cells

To detect the response of deprotected MASTER-compounds in permeabilized Jurkat T cells, approx. $3.5 \times 10^7$ cells per experiment were transferred into a 50 mL falcon tube and centrifuged for 5 min at 300 x g and room temperature. After removal of the supernatant using a suction pump equipped with a glass capillary, the cell pellet was rinsed twice with 10 mL preparation buffer (120 mM KCl, 10 mM NaCl, 1.2 mM $MgCl_2$, 0.533 mM $CaCl_2$, 1 mM EGTA, 10 mM HEPES, pH 7.2 set with 7 M KOH) and resuspended in saponin (80 µg/mL diluted in

preparation buffer) for 10 min in a water bath at 37 °C. After incubation, the cells were centrifuged (5 min at 300 x $g$, room temperature) and the supernatant was carefully removed using the suction pump. The permeabilized cells were rinsed once in 10 mL intracellular buffer (120 mM KCl, 10 mM NaCl, 1.2 mM MgCl$_2$, 10 mM HEPES, pH 7.2 set with 7 M KOH) and resuspended in a final volume of 1 mL intracellular buffer. Next, cells were transferred into a clean quartz cuvette, a small magnetic stirring bar was added and the cuvette was placed in the fluorimeter. Stirring was set to the slowest option. For the measurements, fluorimeter parameters were set to 488 nm excitation and emissions at 525 nm and 650 nm. The acquisition rate was set to 0.5 s$^{-1}$. Before starting the measurement, 25 µg/mL creatine kinase, 200 mM creatine phosphate and 1 µg/mL CalRed 525/650 were added. After the addition of CalRed, data acquisition was started and 1 mM ATP was added after a stable baseline was recorded for at least 1 min. Approximately 5 min after ATP addition 10 µM of a reversible and non-competitive glucose-6-phosphate dehydrogenase (G6PD) inhibitor (g6pdi-1) was added and after a stable baseline was recorded for 2 min, 15 µg recombinant human HN1L/JPT2 was added. Further additions were performed by injecting the compounds through a membrane in the closed lid using a needle-equipped gastight syringe.

The measurement was calibrated by stepwise injecting solutions of ionomycin (1 µM final concentration), CaCl$_2$ (2 mM final concentration), and Tris-EGTA (60 mM final concentration of Tris-base and 8 mM EGTA). For data analysis, the ratio of 525 nm emission divided by 650 nm emission for each timepoint was calculated. Ratio values were converted to [Ca$^{2+}$] concentrations based on Grynkiewicz et al.[69] using a K$_d$ for CalRed of 330 nM. For every injection, the mean Ca$^{2+}$ concentration over 10 s before the addition was subtracted from the mean Ca$^{2+}$ concentration over the following 10 s after the injection and plotted against the mean Ca$^{2+}$ concentration over the 10 s before the injection. The resulting datasets were analyzed using linear regression.

**Planar lipd bilayer measurements**

RYR1 was purified from rabbit skeletal muscles. Purified RYR1 was reconstituted into liposomes as described before ref. [71]. Purified RYR1 was concentrated to <0.5 mL and incubated at a molar ratio 1:1,000 with phosphatidylethanolamine and phosphatidylcholine (5:3 ratio) in 10% CHAPS for 10 min prior to loading onto a hand-packed G50 column pre-equilibrated with buffer S containing 300 mM NaCl,10 mM HEPES (pH 7.4), 1 mM EGTA, 0.5 mM tris(2-carboxyethyl)phosphine. Liposome formation and RYR1 incorporation occurred on the column and proteoliposomal RYR1 was eluted at 0.20–0.25 mL/min in buffer S, frozen in liquid nitrogen, and stored at −80 °C.

Proteoliposomal RYR1 was fused to planar lipid bilayers formed by painting a lipid mixture of phosphatidylethanolamine and phosphatidylcholine (Avanti Polar Lipids) in a 5:3 ratio in decane (Sigma) across a 200 µm hole in polysulfonate cups (Warner Instruments) separating two chambers. The trans chamber (1.0 mL), representing the intra-SR (luminal) compartment, was connected to the head stage input of a bilayer voltage clamp amplifier. The cis chamber (1.0 mL), representing the cytoplasmic compartment, was held at virtual ground. Symmetrical solutions used were as follows: 1 mM EGTA, 250/125 mM HEPES/ Tris, 50 mM KCl (pH 7.35) as *cis* solution and 53 mM Ca(OH)$_2$, 50 mM KCl, 250 mM HEPES (pH 7.35) as *trans* solution. The concentration of free Ca$^{2+}$ in the cis chamber was calculated with WinMaxC program. Proteoliposomal RYR1was added to the *cis* side. Then 400–500 mM KCl was added to generate the driving force and induce lipid bilayer fusion. The *cis* chamber was perfused with *cis* solution to remove KCl after RYR1 incorporation. For all recordings, 1 µM free Ca$^{2+}$ was first added to the *cis* side for the 3 min baseline recordings. Then, for the NADP group, 100 nM deMASTER-NADP and 10 µM purified HN1L/JPT2 was added together and recorded for at least 3 min. For the NAADP group, 100 nM deMASTER-NAADP and 10 µM purified HN1L/JPT2 were incubated sequentially and recorded for at least 3 min for each step.

Single-channel currents were recorded at 0 mV using a Bilayer Clamp BC-525C (Warner Instruments), filtered at 1 kHz using a Low-Pass Bessel Filter 8 Pole (Warner Instruments), and digitized at 4 kHz. All experiments were performed at room temperature (23 °C). Data acquisition was performed using Digidata 1322 A and Axoscope 10.1 software (Axon Instruments). The recordings were analyzed using Clampfit 10.1 (Molecular Devices) and imaged by Graphpad Prism software. Each datapoint was normalized based on the baseline open probability (Po of 1 µM Ca).

**Preparation of *Hn1l/Jpt2*$^{-/-}$ Neuro2A cells**

*Hn1l/Jpt2*$^{-/-}$ cell lines were developed by generating frameshift mutations by targeting exon2 (sgRNA#2 TTCAAGCAAGCCTAATAGGA) of the *Hn1l/Jpt2* gene using the CRISPR-Cas9 technology. A detailed, sequential step-by-step protocol is described in ref. [72]. In brief, the sgRNA was cloned into pSpCas9(BB)-2A-GFP (PX458). Neuro2a (N2a) cells (obtained from the DSMZ German Collection of Microorganisms and Cell Cultures GmbH) were transfected, and GFP-expressing cells were isolated by FACS sorting. In the obtained individual cell clones, putative null alleles were identified by Sanger sequencing with the aid of the designated algorithm"Interference of CRISPR Edits" (ICE)[73]. PCR amplicons covering the mutated site of high potential knockout clones (according to ICE scores) were amplified and NGS sequenced ( > 33.143 reads, Amplicon-EZ service at GeneWiz, Azenta Life Sciences). In clones 1G4 and 1F3 we identified frameshift mutations on all alleles. Clone 1G4: Δ+1 allele c.G203Δ + 1 AfsX214 with insertion of Δ+1(A) at position 203 of the transcript (ENSMUST00000024981.9) identified in 49,57% of 33.143 reads leading to a preliminary stop codon at position 46 of protein sequence (p.R42KfsX46), and a Δ+2 allele c.G203Δ +2TAfsX331 with insertion of Δ+2(TA) at position 203 of the transcript in 49.40% of all reads leading to a preliminary stop codon at position 85 of protein sequence, (p.R42IfsX85). Clone 1F3: a Δ+1 allele c.G203Δ +1AfsX214 with insertion of Δ+1A at position 203 of the transcript was identified in 98.72% of 130.074 reads leading to to a preliminary stop codon at position 46 of protein sequence (p.R42KfsX46). Lack of HN1L/JPT2 protein expression was confirmed by SDS-PAGE/western blot (detail see below). *Hn1l/Jpt2*$^{-/-}$ clones 1G4 and 1F3 were then used for Ca$^{2+}$ imaging experiments with respect to analysis of MASTER-NAADP mediated Ca$^{2+}$ signaling.

**SDS-PAGE and western blot of HN1L/JPT2**

For the preparation of the S10 cytosolic fraction, a suspension of 10 to 30 million WT Jurkat, WT KHYG-1, or Neuro2a (WT and *Hn1l/Jpt2*$^{-/-}$) cells were centrifuged (518 $g$, 5 min, room temperature), followed by resuspension of the cell pellet in Dulbecco's phosphate-buffered saline without Ca$^{2+}$ and Mg$^{2+}$ (pH 7.4) with protease inhibitor (cOmplete protease inhibitor cocktail EDTA free, Roche). Next, using Sonopuls GM 70 (Bandelin Electronic) the suspension was ultrasonicated for 30 s on ice to lyse the cells. This procedure was repeated 3 times. Cell lysate was then centrifuged (518 $g$, 5 min, 4 °C) to pellet cell debris. The supernatant was removed and subjected to another centrifugation step (6000 $g$, 10 min, 4 °C), and the resulting supernatant was collected for further centrifugation (10,000 $g$, 30 min, 4 °C), which finally yielded the S10 fraction. Bradford assay and Nanodrop was used to measure the protein concentration. Until use, the S10 fraction was stored at −20 °C with protease inhibitor. Protein samples were separated by sodium dodecyl sulfate polyacrylamide gel electrophoresis (SDS-PAGE). Therefore, samples were pre-mixed with 4x Laemmli buffer [73.8% (v/v) 6x SDS-PAGE loading buffer stock solution, 21.1% (v/ v) 20% SDS solution, and 5.1% (v/v) β-mercaptoethanol], then samples were boiled at 95 °C for 5 min to ensure denaturation of the proteins and either 12.5% precast gels (Bio-Rad Laboratories) or 15% self-cast gels were used for separation. SDS-PAGE was run for 10 min at 90 V to ensure proper loading of protein samples, continued by separating for 40–45 min at 180 V. Next, western blotting experiments were

performed. For this purpose, the separated proteins were transferred from the gel onto a 0.45 μm pore-sized polyvinylidene difluoride (PVDF) membrane (Merck Millipore Ltd.) by either tank electroblotting (at constant current of 200 mA for 90 min) or semi-dry blotting (constant current of 200 mA for 20 min). For detection of HN1L/JPT2, PVDF-membranes were incubated overnight at 4 °C with a primary anti-HN1L antibody (1:1000, orb1412, Biorbyt or 1:2000, HPA041888, Atlas Antibodies). PVDF-membranes were then rinsed, following incubation for 1 h at RT with an anti-rabbit secondary antibody conjugated with horseradish peroxidase (1:5000, ab6721, Abcam) or peroxidase AffiniPure goat anti-rabbit immunoglobulin G (1:20.000, 111-035-045, Jackson ImmunoResearch Europe LTD.). Incubation of PVDF-membranes with the chemiluminescent substrate Super Signal West Pico (Thermo Fisher Scientific, Waltham, Massachusetts, USA) resulted in chemiluminescence catalyzed by HRP. Chemiluminescence of the blot was detected using the fusion FX edge system (Vilber). After the detection of HN1L/JTP2, the PVDF-membranes were stripped by 30 min incubation with 30% (v/v) $H_2O_2$ at RT with constant shaking, followed by 6 times washing with TBS-T (Tris-buffered saline with 0.05% (v/v) Tween20). As loading control, actin was stained with anti-α-actin (1:500, MAB1501, Merck) or anti-β-actin (1:500, SC-47778, Santa Cruz).

## Statistical analysis

Prism 10 (GraphPad Software, USA) and Excel (Microsoft, USA) were used for data analysis. All data are shown as mean ± SEM of at least three independent experiments. Datasets were first tested for normal distribution using the Kolmogorov-Smirnov test. The none-normally distributed data sets were further tested using a non-parametric Mann-Whitney U test or a Kruskal-Wallis test (one-way ANOVA) with subsequent Dunn's correction for multiple testing. Normally distributed data sets were tested using ordinary one-way ANOVA with Tukey's multiple comparisons test or ANOVA mixed effects test with Holm-Šídák correction for multiple testing as indicated. Linear regression was performed using the "simple linear regression" function in Prism 10 and slope deviation from zero was analyzed using a F test. Differences with a $P$ value of <0.05 were considered statistically significant: $*P < 0.05$; $**P < 0.005$; $***P < 0.001$; $****P < 0.0001$.

## Reporting summary

Further information on research design is available in the Nature Portfolio Reporting Summary linked to this article.

## Data availability

The data supporting the findings of this study are available within the paper and its Supplementary Information and Source Data files. Large high-resolution imaging raw data files that were generated and used in this study are available from the corresponding author upon request. Source data are provided with this paper.

## Code availability

A Python implementation of the $Ca^{2+}$ image postprocessing and microdomain analysis routines is publicly available (license: Apache 2.0) at Kovacevic D, Woelk L-M, Husseini H, Förster F, Werner R. IPMI-ICNS-UKE/DARTS: DARTS FIMMU 2024 release (Version 1.0.0). Zenodo entry 10459242. The corresponding documentation can be found in ref. 51.

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

## Acknowledgements

This study was supported by grants from the Deutsche Forschungsgemeinschaft (DFG) (project number 335447717; SFB1328, project A01 to AHG, project A02 to B-PD and RW, project A03 to SH, project A04 to CM, project A21 to MF), SFB 1118 (project number 236360313), project S03 to RM and MF, and DFG project 516286863 to B-PD and MA. KJW is supported by SFB1328 and funded by Jung-Stiftung für Wissenschaft und Forschung, Hamburg. We are grateful to Prof. Aymelt Itzen (Dept. of Biochemistry and Signal Transduction, UKE, Hamburg, Germany) who provided extracted plasmids pETM33 with His-tag and GST-tag.

## Author contributions

Conceptualization: A.H.G., C.M., M.W., B.-P.D., M.F., O.B.C., S.H., M.A., R.W. Methodology: B.-P.D., M.W., S.K., F.M., R.W., V.T., R.M., L.-M.W., T.B., F.F., K.J.W., L.H., M.N. Investigation: S.K., F.M., M.W., P.D., M.H., F.G., K.J.W., D.K., I.K., B.T.K., M.H., V.G., L.H., Visualization: S.K., F.M., M.W., K.J.W., D.K., V.T., L.H., B.-P.D., C.M., A.H.G. Funding acquisition: A.H.G., S.H., R.W., B.-P.D., M.W., M.A., M.F., C.M. Project administration: A.H.G., M.W., B.-P.D., C.M., M.F. Supervision: A.H.G., S.H., R.W., B.-P.D., M.W., M.A., M.F., C.M., O.B.C. Writing – original draft: A.H.G., S.K., F.M., S.H., R.W., B.-P.D., M.W., M.F., C.M. Writing – review & editing: all authors.

## Funding

## Competing interests

The authors declare no competing interests.
