## [Peer Review File · Nature Communications]

MASTER-NAADP: a membrane permeable precursor of the Ca²⁺ mobilizing second messenger NAADPREVIEWER COMMENTS

Reviewer #1 (Remarks to the Author):

I saw this work at conference and am very impressed by it. The level of detail in synthesis and use of NAADP-mimetic probes is greatly appreciated. The effects of the synthesized compounds in multiple cellular models appears to be robust and accordingly, I think this body of work merits publication in Nature Communication.

There's one problem with the work, however, that needs to be corrected. In lines 43 to 49, the authors review how NAADP is formed. They completely ignore a fully described mechanism by which an IL8 signal is propagated to drive Cx43 to pump NADP into the lumen of end-lysosomes where a specific membrane bound form of CD38 is colocalized with enzymes that form NAAD. Thus, in a signal and pH-dependent manner, NAADP is formed. Though Nam et al was cited, this NAADP formation mechanism was not discussed as such and this does a disservice to the field. Thus, the authors are requested to move a discussion of ref 60 into the introductory paragraph.

Reviewer #2 (Remarks to the Author):

This manuscript is a straightforward and thorough description of the development of a cell-permeable NAADP analog. This molecule would facilitate studies of the complicated NAADP signaling pathway. This should be of importance to the field, although it has a rather difficult, multi-step synthesis. The target compounds and their synthetic intermediates were well characterized using ¹H-NMR, ¹³C-NMR, ³¹P-NMR, and mass spectrometry. Yields for reactions were reported. The analog is demonstrated to be active in several cell types. A comparison is made to another cell-permeable NAADP analog, NAADP-AM. The following items should be modified to improve the usefulness of the publication.

1. Line 51 add "as demonstrated by Lin-Moshier et al. (JBC 287 (4) 2296, 2012)" and hypothesized... The demonstration was published prior to the hypothesis.
2. The MASTER-NAADP and MASTER-NADP are hydrolyzed to produce a novel NAADP analog or a novel NADP analog. The analogs differ at several points from the natural molecules. Therefore, the analogs should be tested by microinjection to determine their dose-response as free analogs compared to NAADP.
3. The liberated NAADP and NADP analogs are not direct comparisons. The NADP analog lacks the 3' fluorine that is present in the NAADP in order to prevent migration of the 2' phosphate group. This migration could happen in the NADP analog. It would be best if the NAADP and NADP analogs only differ at the carboxylate group.
4. NAADP has a characteristic dose-response curve in eukaryotic cells, with inhibition at higher concentrations. Only one dose is shown for each cell type. A dose-response should be shown for treatment of at least one cell type to confirm a similar shape of dose-response as that of NAADP.
5. The article mentions both JPT2/HN1L and Lsm12 as NAADP-binding proteins. HN1L knockout cells lose responsiveness to MASTER-NAADP, yet they still supposedly express Lsm12. Does the released NAADP analog not bind to Lsm12? The researchers should test Lsm12 and double knockout cells in addition to HN1L knockout cells.

6. Several of the intermediates in the syntheses, for example **36**, have been previously described. The authors should cite references for each of the previously published compounds in their descriptions.

7. The authors should show data for release of the NADP analog from MASTER-NADP as they have shown for MASTER-NAADP in Figure 2B. This may be added to the supplementary material.

Minor changes

1. Line 40 add "an" impurity
2. Line 120 "off" not of
3. Line 351 sentence needs to be rewritten
4. Line 367 "and" not und
5. Line 1037 "western blot"
6. Line 1045 Provide source of KHYG-1 cell line
7. Line 1048 Provide source of Neuro2a cells
8. Line 1052 "Jurkat"
9. Line 1053 "split" not splitted

Reviewer #3 (Remarks to the Author):

In this paper, Krukenberg et al. have synthesized and tested a membrane permeable stable mimetic of calcium mobilizing second messenger NAADP. The authors designed a precursor compound named as MASTER-NAADP, which upon cleavage by intracellular esterases, releases a close derivative of NAADP, benzoic acid C-nucleoside, 3'F-adenosine-diphosphate. As a control, they also designed a compound called MASTER-NADP, which releases an NAD derivative and does not induce intracellular calcium flux to the same extent as NAADP, in a given cell line. The authors generate and use Hn1l-/- cell lines to demonstrate the specificity of their compound and use a variety of cell lines to show the calcium microdomains elicited and propose that a common pathway is being targeted by this compound across different cell lines. I have the following comments.

1. Figure 3 general: A detailed protocol for calcium microdomain imaging needs to be described in the methods section. Merely referring to an older paper is not sufficient. It is unclear what is actually defined as a calcium microdomain and the respective single and noise for each cell type? Is there a definite area, pixel size, a specific intensity that was used to describe a region as a calcium microdomain and was it kept constant in primary T cells versus Jurkat or Neuro2A cells? Were all frames used to quantify and plot the number of microdomains in each figure or only those close to the PM? How did the authors ensure that the frame being analyzed was comparable across all groups, for instance, in terms of distance from the coverslip or the top of the cell?

2. Figure 3 general: In line 322, the authors state that they detect calcium microdomains close to the plasma membrane in lymphocytes. It is unclear what distance range/ frames were chosen and considered as close to plasma membrane (PM). Lymphocytes, in general have a relatively large nucleus when compared to their overall cell size. The distance from

plasma membrane to the nucleus is not much. How many total frames were considered close to PM and analyzed in a Jurkat T lymphocyte versus murine T lymphocyte or an NK cell versus Neuro 2A cells, respectively, between the nucleus and the PM.

3. Figure 3 C,E,G,I: From the bar plots that are shown in 3C,E,G,I, it is unclear how the number of microdomains will shift or change with time beyond 15s? A frequency versus time period graph showing the dynamics of calcium microdomains over an extended time period (entire 2 minutes) would probably be more useful here. For instance, what may seem as a microdomain at 7.5s will likely disappear or diffuse or become more intense at 15sec or beyond. It is not clear how the authors resolved between these three possibilities in the time period spanning 15sec and beyond.

4. Figure 3F and 3H: The concentration of MASTER-NAADP used in NK cells and Neuro2A cells 1000-fold higher (100uM) than what is used for Jurkat T cells and primary T cells? The biological explanation given in line 320 for such a broad range appears unconvincing. It is hard to envisage a 1000-fold difference in the concentration of cellular esterases between different cell types. The authors should use 100uM concentration of MASTER-NAADP and MASTER-NAD to stimulate Jurkat T whether the difference is still intact. Given that the difference in Ca peaks number is only 2-fold (Figure 3C), the differences in stability/ bioavailability of these two compounds can easily account for this kind of difference.

5. Figure 3G: The number of calcium peaks/ frame seen in response to 100uM MASTER-NAD is similar to what is seen in Jurkat cells with 100nM MASTER-NAADP (Figure 3C). It is unclear whether this is due to high background calcium signal in NK cells? Or do NK cells respond to MASTER-NAD at 100uM concentration, unlike other cells? Perhaps using another control will help clarify this point since the concentration of MASTER_NAADP used in NK cells to elicit a response over and above MASTER-NAD is 100uM.

6. Many pharmacological agents can have variable effects over different concentration ranges either due to different modes of binding or non-specific binding. For instance, 2-APB inhibits as well as potentiates CRAC currents at two different concentrations. The authors should describe in detail how non-specific binding effects were ruled out for a concentration range spanning 100nM to 100uM. This should be done for both MASTER-NAADP and MASTER-NAD.

7. Figure 4: The authors should mention what imaging speed was used for global calcium imaging. If it is comparable to their calcium microdomain protocol and performed at 40 frames (image pair)/ sec there will be significant bleaching of Fura-2 especially at later time points and could result in underestimation of peaks. If it is not performed at the 40 frames (image pair)/ sec speed and is closer to the usual 1 image pair every 2-4 sec, how do the authors reconcile the loss of several peaks in the intervening period between each image pair with their data in Figure 3?

8. Figure 4D, why does the mean max amplitude appear very different between panel A and B but quite similar in panel D.

9. Figure 5: It is unclear how the analysis of NAADP-AM helps in making the central point in the paper and why did the authors not choose 100uM concentration here?

10. Figure 6: It is intriguing to see that the global response of KHYG-1 and Neuro 2A cells is so different from Jurkat cells in the max amplitude as well as %age of responding cells. Of course the concentration of MASTER-NAADP used is 1000-fold different. However, another possibility is that the levels of Hn1l expression is different across these three cell lines. If the

expression of the receptor is indeed 1000-fold different in different cell lines, it needs to be shown by using quantitative analysis of Hn1l levels. Another difference in Figure 4 and 6 is the presence of Synta66 in Figure 4. Why did the authors choose to omit Synta 66 in Figure 6? Wouldn't it be more important in Figure 6 where 100uM of MASTER-NAADP is being used?

11. Figure 3,4,6: Why is the %age of cells responding to MASTER-NAADP only ~15% in Jurkat and KHYG-1? It is ~80% in Neuro2A cells. The authors should resolve this difference in response and perform downstream signaling analysis on a population of cells as suggested below to allow assessment of physiological effects of the pro-drug MASTER-NAADP.

General comments:

1. A binding assay between Hn1l and MASTER-NAADP would help further establish the specificity of the MASTER-NAADP.
2. It will strengthen the study if downstream signaling analysis was performed. It will further establish the specificity of MASTER-NAADP.
3. Since authors have called MASTER-NAADP a pro-drug, it will be nice to rule out toxicity over prolonged periods. In vivo administration and downstream gene expression analysis would help the study further.

Reviewer #4 (Remarks to the Author):

The authors have presented the design, synthesis and testing of a bis-prodrug of benzoic acid modified C-nucleoside 3'-F-adenosine diphosphate for its ability to stimulate NAADP signaling. In that regard, MASTER-NAADP, a Membrane permeable, Stabilized, bio-reversibly protected prodrug of NAADP was pursued. This process is important as NAADP acts as a second messenger and stimulates Ca²⁺ signaling in many different cell types. The paper is quite well written other than some few typos/grammatical errors below. Interestingly, the chemistry proved to be rather challenging but was ultimately achieved. Moreover, their findings are likely to be of strong interest to the field, due to the development of this useful tool to study NAADP signaling and downstream events. Other than the minor issues below, the paper is ready for publication as it stands.

page 2, line 54: define ER

page 3, line 82: insert "the" before 1,2-position

page 4, line 156: delete also before neuronal

page 5, line 170: insert "a" before few

page 5, line 184: insert "the" before mean

page 5-6: why is figure 5 discussed before figure 4 (i.e. use numerical order)

page 5, line 196: change digest to digestion

page 7, line 251: removed also before "the esterification"

page 10, line 345: should this just say Methods, not online methods?

Re-submission of revised manuscript NCOMMS-23-16636-T

Point-by-point reply to REVIEWER COMMENTS

Reviewer #1 (Remarks to the Author):

I saw this work at conference and am very impressed by it. The level of detail in synthesis and use of NAADP-mimetic probes is greatly appreciated. The effects of the synthesized compounds in multiple cellular models appears to be robust and accordingly, I think this body of work merits publication in Nature Communication.

Reply: We are happy that reviewer 1 states that the manuscript should be published in Nature Communications.

There's one problem with the work, however, that needs to be corrected. In lines 43 to 49, the authors review how NAADP is formed. They completely ignore a fully described mechanism by which an IL8 signal is propagated to drive Cx43 to pump NADP into the lumen of end-lysosomes where a specific membrane bound form of CD38 is colocalized with enzymes that form NAAD. Thus, in a signal and pH-dependent manner, NAADP is formed. Though Nam et al was cited, this NAADP formation mechanism was not discussed as such and this does a disservice to the field. Thus, the authors are requested to move a discussion of ref 60 into the introductory paragraph.

Reply: In the revised version we moved the mentioned reference (Nam TS, Park DR, Rah SY, Woo TG, Chung HT, Brenner C, Kim UH. Interleukin-8 drives CD38 to form NAADP from NADP⁺ and NAAD in the endolysosomes to mobilize Ca²⁺ and effect cell migration. FASEB J. 2020 Sep;34(9):12565-12576.9) to the introduction and highlight this finding by Profs Brenner and Kim.

Reviewer #2 (Remarks to the Author):

This manuscript is a straightforward and thorough description of the development of a cell-permeable NAADP analog. This molecule would facilitate studies of the complicated NAADP signaling pathway. This should be of importance to the field, although it has a rather difficult, multi-step synthesis. The target compounds and their synthetic intermediates were well characterized using ¹H-NMR, ¹³C-NMR, ³¹P-NMR, and mass spectrometry. Yields for reactions were reported. The analog is demonstrated to be active in several cell types. A comparison is made to another cell-permeable NAADP analog, NAADP-AM. The following items should be modified to improve the usefulness of the publication.

Reply: We are glad that our work is considered useful for the NAADP field.

1. Line 51 add "as demonstrated by Lin-Moshier et al. (JBC 287 (4) 2296, 2012)" and hypothesized... The demonstration was published prior to the hypothesis.

Reply: Revised as suggest by reviewer 2.

2. The MASTER-NAADP and MASTER-NADP are hydrolyzed to produce a novel NAADP analog or a novel NADP analog. The analogs differ at several points from the natural molecules. Therefore, the analogs should be tested by microinjection to determine their dose-response as free analogs compared to NAADP.

Reply: Deprotected MASTER-NAADP, benzoic acid C-nucleoside, 2'-phospho-3'-F-adenosine-diphosphate, and deprotected MASTER-NADP, benzamide C-nucleoside, phospho-adenosine-diphosphate, were both analyzed in 2 systems, (i) permeabilized Jurkat T cells for Ca²⁺ release, and (ii) channel open probability of RYR1 fused into lipid planar bilayers. Data are shown in **new** Fig 3 and described on p. 5. Of note, in both systems the addition of exogenous HN1L/JPT2 is required to observe Ca²⁺ release (Fig 3 A-D) or increased open probability of RYR1 (Fig 3 E,F).

These data clearly demonstrate that benzoic acid C-nucleoside, 2'-phospho-3'-F-adenosine-diphosphate is a *bona fide* NAADP mimic, whereas benzamide C-nucleoside, phospho-adenosine-

diphosphate is a suitable control for potential unspecific effects.

3. The liberated NAADP and NADP analogs are not direct comparisons. The NADP analog lacks the 3' fluorine that is present in the NAADP in order to prevent migration of the 2' phosphate group. This migration could happen in the NADP analog. It would be best if the NAADP and NADP analogs only differ at the carboxylate group.

Reply: In fact, deprotected MASTER-NAADP, benzoic acid C-nucleoside, 2'-phospho-3'-F-adenosine-diphosphate, is a single compound (Fig 2 B), whereas deprotected MASTER-NADP, benzamide C-nucleoside, phospho-adenosine-diphosphate, is a mixture of the 2'- and 3'-phosphate isomers (**new** suppl Fig S3). We agree with reviewer 2 that it would be of advantage to have the 3'-fluorine derivative also for the control compound. However, it was impossible to synthesize this molecule due to stability reasons, thereby largely limiting its usefulness as control compound for the community. Despite the presence of the 2 isomers of benzamide C-nucleoside, phospho-adenosine-diphosphate, the lack of the benzoic acid group is sufficient to prevent any activity towards HN1L/JPT2-mediated Ca^{2+} signaling effects, as demonstrated in Fig 3.

4. NAADP has a characteristic dose-response curve in eukaryotic cells, with inhibition at higher concentrations. Only one dose is shown for each cell type. A dose-response should be shown for treatment of at least one cell type to confirm a similar shape of dose-response as that of NAADP.

Reply: We agree with reviewer 2 that it is an interesting question whether the bell-shaped concentration-response curve known for NAADP would also be seen with MASTER-NAADP. In Fig 5 and suppl Fig S8, we used 100 nM, 10 μ M, or 100 μ M MASTER-NAADP or MASTER-NADP in Jurkat T cells. At least the mean number of specific Ca^{2+} peaks decreased at 10 μ M, or 100 μ M when compared to 100 nM (the 'mean number of specific Ca^{2+} peaks' is calculated by subtracting the values for MASTER-NADP from the values for MASTER-NAADP; calculated from data shown in Fig 5D and suppl Fig S8C are 0.361 (100 nM), 0.140 (10 μ M), or 0.183 (100 μ M)). These data also demonstrate how important a proper control (MASTER-NADP) is to achieve a meaningful interpretation of the data. Especially for 100 μ M

5. The article mentions both JPT2/HN1L and Lsm12 as NAADP-binding proteins. HN1L knockout cells lose responsiveness to MASTER-NAADP, yet they still supposedly express Lsm12. Does the released NAADP analog not bind to Lsm12? The researchers should test Lsm12 and double knockout cells in addition to HN1L knockout cells.

Reply: Again a very interesting point raised. When looking at experiments that compare WT to *Hn1l/Jpt2^{-/-}* cells, effects of MASTER-NAADP in *Hn1l/Jpt2^{-/-}* cells are either not significantly different from background controls or extremely low as compared to the values obtained with MASTER-NAADP. That is true for local Ca^{2+} microdomains (Figs 4C, 4I, S4, S5) or for global Ca^{2+} signaling (Figs 5F vs 5G, 6H-K). Given these clear results, namely (almost) full abolishment of Ca^{2+} signaling evoked by MASTER-NAADP when compared to MASTER-NADP or background controls, there would be obviously no or very modest effects in Lsm12 knock-outs. Thus, we will certainly look for these effects in the future, but for the moment they are out of the scope of our manuscript.

6. Several of the intermediates in the syntheses, for example **36**, have been previously described. The authors should cite references for each of the previously published compounds in their descriptions.

Reply: References for previously described compounds are

44. Markiewicz, W. T. & Wiewiórowski, M. A new type of silyl protecting groups in nucleoside chemistry. *Nucleic Acids Res* **5**, s185–s190 (1978).

45. Zhu, X.-F., Williams, H. J. & Scott, A. I. Aqueous trifluoroacetic acid—an efficient reagent for exclusively cleaving the 5'-end of 3',5'-TIPDS protected ribonucleosides. *Tetrahedron Letters* **41**, 9541–9545 (2000).

47. Wender, P. A. *et al.* The Pinene Path to Taxanes. 6. A Concise Stereocontrolled Synthesis of Taxol. *J. Am. Chem. Soc.* **119**, 2757–2758 (1997).

7. The authors should show data for release of the NADP analog from MASTER-NADP as they have shown for MASTER-NAADP in Figure 2B. This may be added to the supplementary material.

Reply: HPLC analysis of esterase digest of MASTER-NADP to benzamide C-nucleoside, phospho-adenosine-diphosphate is now shown in suppl Fig S3.

Minor changes

1. Line 40 add “an” impurity
2. Line 120 ‘off” not of
3. Line 351 sentence needs to be rewritten
4. Line 367 “and” not und
5. Line 1037 “western blot”
6. Line 1045 Provide source of KHYG-1 cell line
7. Line 1048 Provide source of Neuro2a cells
8. Line 1052 “Jurkat”
9. Line 1053 “split” not splitted

Reply: All minor points were revised as suggested.

Reviewer #3 (Remarks to the Author):

In this paper, Krukenberg et al. have synthesized and tested a membrane permeable stable mimetic of calcium mobilizing second messenger NAADP. The authors designed a precursor compound named as MASTER-NAADP, which upon cleavage by intracellular esterases, releases a close derivative of NAADP, benzoic acid C-nucleoside, 3’F-adenosine-diphosphate. As a control, they also designed a compound called MASTER-NADP, which releases an NAD derivative and does not induce intracellular calcium flux to the same extent as NAADP, in a given cell line. The authors generate and use Hn11-/- cell lines to demonstrate the specificity of their compound and use a variety of cell lines to show the calcium microdomains elicited and propose that a common pathway is being targeted by this compound across different cell lines. I have the following comments.

1. Figure 3 general: A detailed protocol for calcium microdomain imaging needs to be described in the methods section. Merely referring to an older paper is not sufficient. It is unclear what is actually defined as a calcium microdomain and the respective single and noise for each cell type? Is there a definite area, pixel size, a specific intensity that was used to describe a region as a calcium microdomain and was it kept constant in primary T cells versus Jurkat or Neuro2A cells? Were all frames used to quantify and plot the number of microdomains in each figure or only those close to the PM? How did the authors ensure that the frame being analyzed was comparable across all groups, for instance, in terms of distance from the coverslip or the top of the cell?

Reply: A detailed protocol of Ca²⁺ microdomain imaging is now included in the Methods section (p.33/34). Subcellular Ca²⁺ microdomains are defined as small, compact connected sets of pixels with high [Ca²⁺]_i values. To discriminate between signal and noise, noise was determined in a cell-free system in EGTA buffer ([Ca²⁺] = 100 nM) to calculate a threshold for the Ca²⁺ microdomain detection. The threshold for Jurkat cells is 72 nM and for primary T cells, Neuro2A cells and KHYG-1 cells is 112.5 nM. Next, on a frame-by-frame basis the frame-specific mean [Ca²⁺]_i value was calculated and a Ca²⁺ microdomain is detected if (i) its mean [Ca²⁺]_i value is increased above the respective amplitude threshold, (ii) the Ca²⁺ microdomain consists of 6 pixel to 20 pixel. In this manuscript, each whole confocal frame was analyzed for Ca²⁺ microdomains. Concerning the position at the z-axis of the confocal frame used, all cells were focused to the highest possible diameter. This ensures that the z-frame is always located at the center in between coverslip and top of the cell.

2. Figure 3 general: In line 322, the authors state that they detect calcium microdomains close to the plasma membrane in lymphocytes. It is unclear what distance range/ frames were chosen and

considered as close to plasma membrane (PM). Lymphocytes, in general have a relatively large nucleus when compared to their overall cell size. The distance from plasma membrane to the nucleus is not much. How many total frames were considered close to PM and analyzed in a Jurkat T lymphocyte versus murine T lymphocyte or an NK cell versus Neuro 2A cells, respectively, between the nucleus and the PM.

Reply: We carefully re-analyzed the spatio-temporal distribution of Ca^{2+} microdomains evoked by MASTER-NAADP in Jurkat T cells and KHYG-1 cells over the first 15 seconds (new suppl Fig S5 and S7). We used the so-called dartboard representation where cells are shape-normalized and then the amplitudes of aggregated mean Ca^{2+} microdomains from all cells analyzed under a certain condition are plotted segment-wise in red color intensity (for reference see Woelk, L.-M. *et al.* DARTS: an open-source Python pipeline for Ca^{2+} microdomain analysis in live cell imaging data. *Frontiers in Immunology* **14**, (2024); which is cited as no. 51 in the revised version). Further, for kinetic analyses we subtracted segment-wise the intensity data for each of the 15s time points of MASTER-NADP from the respective values for MASTER-NAADP to obtain the specific data. These subtraction plots are now included in new suppl Fig S5B and S7B. For Jurkat T cells, initial signals induced by MASTER-NAADP were observed in the center of the cells (suppl Fig S5B, 1-3s), but rapidly spread to more peripheral parts of the cells (suppl Fig S5B, 4-6s), followed by a phase of more even distribution of the Ca^{2+} microdomains throughout the cells (suppl Fig S5B, 7-11s). For KHYG-1 cells, the corresponding subtraction plots demonstrate initial Ca^{2+} microdomains confined to a ring-like structure in the cytosol (suppl Fig S7B, 1-4s). Then, Ca^{2+} microdomains were also observed close to the plasma membrane and in the inner part of the cell (suppl Fig S7B, 5-9s), followed by a more even distribution of Ca^{2+} microdomains across the whole cell (suppl Fig S7B, 10-15s). These refined analyses are also described in the revised manuscript.

3. Figure 3 C,E,G,I: From the bar plots that are shown in 3C,E,G,I, it is unclear how the number of microdomains will shift or change with time beyond 15s? A frequency versus time period graph showing the dynamics of calcium microdomains over an extended time period (entire 2 minutes) would probably be more useful here. For instance, what may seem as a microdomain at 7.5s will likely disappear or diffuse or become more intense at 15sec or beyond. It is not clear how the authors resolved between these three possibilities in the time period spanning 15sec and beyond.

Reply: There are two limitations for doing analysis much longer than 15s: (i) photobleaching of the Ca^{2+} dyes, and (ii) merging of local Ca^{2+} signals. Photobleaching is due to the fact that we acquire 40 frames per second resulting in photochemical decay of the Ca^{2+} dyes. In our initial paper on Ca^{2+} microdomains, a major part was devoted to selection of the best Ca^{2+} dyes (Wolf IM, Diercks BP, Gattkowsky E, Czarniak F, Kempinski J, Werner R, Schetelig D, Mittrücker HW, Schumacher V, von Osten M, Lodygin D, Flügel A, Fliegert R, Guse AH. Frontrunners of T cell activation: Initial, localized Ca^{2+} signals mediated by NAADP and the type 1 ryanodine receptor. *Sci Signal*. 2015 Oct 13;8(398):ra102. doi: 10.1126/scisignal.aab0863. PMID: 26462735). It is very important to use a bright, fast, emission-shift dye, but most of these dyes bleach rapidly, sometimes within 1 s under our experimental conditions. Only the use of Fluo4 and Fura-Red allows to measure for approx. 15s. Nevertheless, we analyzed the data for 25s and show the results in the new suppl Fig S6. Since after 25 s in the majority of cells the local Ca^{2+} microdomains merge into a global Ca^{2+} signal analysis at further time points is not meaningful.

4. Figure 3F and 3H: The concentration of MASTER-NAADP used in NK cells and Neuro2A cells 1000-fold higher (100uM) than what is used for Jurkat T cells and primary T cells? The biological explanation given in line 320 for such a broad range appears unconvincing. It is hard to envisage a 1000-fold difference in the concentration of cellular esterases between different cell types. The authors should use 100uM concentration of MASTER-NAADP and MASTER-NAD to stimulate Jurkat T whether the difference is still intact. Given that the difference in Ca peaks number is only 2-fold (Figure 3C), the differences in stability/ bioavailability of these two compounds can easily account for this kind of difference.

Reply: From the viewpoint of usability of our MASTER-compounds, the major point is to obtain a high difference in NAADP evoked Ca^{2+} signals for MASTER-NAADP vs. MASTER-NADP. The absolute concentration required depends on (i) diffusion rate of the MASTER-compounds through the plasma membrane, (ii) esterase activity present in the cytosol towards the MASTER-compounds, (iii)

degradation of the deprotected NAADP analogue, benzoic acid C-nucleoside, 2'-phospho-3'-adenosine-diphosphate, or of the deprotected NADP analogue, benzamide C-nucleoside, phospho-adenosine diphosphate, and (iv) expression level and activity of xenobiotic efflux pumps. These parameters do vary between different cell types and very likely account for the differences observed between T cells, KHYG-1 cells, or Neuro2A cells. However, determination of all these parameters is out of the scope of the current manuscript.

Nevertheless, by optimization of the absolute concentration of the MASTER compounds used in the cells types included in the manuscript, even with a 1000-fold difference between, e.g. T cells (100 nM) and Neuro2A cells (100 μ M), will guide member of the community to adapt this concentration to many other cell types in the future.

We also demonstrate in the revised version a significant decrease of specificity in T cells at 100 μ M of the MASTER compounds (suppl Fig S8 B-E) as compared to 100 nM (Fig 5).

Our intention is to demonstrate a SPECIFIC Ca^{2+} mobilizing effect in very different cell types to show the usefulness of this pair of compounds, one precursor, MASTER-NAADP, that would release the active NAADP mimic, and the other, MASTER-NADP, to control for unspecific effects. We clearly show for all cell types used, T cells, NK cells, Neuro2A cells, significant differences in Ca^{2+} mobilization by optimal concentrations of MASTER-NAADP vs MASTER-NADP.

5. Figure 3G: The number of calcium peaks/ frame seen in response to 100uM MASTER-NAD is similar to what is seen in Jurkat cells with 100nM MASTER-NAADP (Figure 3C). It is unclear whether this is due to high background calcium signal in NK cells? Or do NK cells respond to MASTER-NAD at 100uM concentration, unlike other cells? Perhaps using another control will help clarify this point since the concentration of MASTER_NAADP used in NK cells to elicit a response over and above MASTER-NAD is 100uM.

Reply: We are very grateful to Reviewer 3 for highlighting this point. During careful re-analysis of the Ca^{2+} microdomain data in KHYG-1 cells, we realized that we used wrong thresholds to detect the initial increase in Ca^{2+} microdomains upon MASTER-compound addition; accordingly the values for both MASTER-NAADP and MASTER-NADP were too high. As can be seen now in the dartboard subtraction plot in suppl Fig S7, this mistake was corrected and the data in new Fig 4 G (previously Fig 3G) are similar to Jurkat T cells (Fig 4C) and Neuro2A cells (Fig 4I).

6. Many pharmacological agents can have variable effects over different concentration ranges either due to different modes of binding or non-specific binding. For instance, 2-APB inhibits as well as potentiates CRAC currents at two different concentrations. The authors should describe in detail how non-specific binding effects were ruled out for a concentration range spanning 100nM to 100uM. This should be done for both MASTER-NAADP and MASTER-NAD.

Reply: The situation described here by reviewer 3 for 2-APB is different from the pair of MASTER compounds. The deprotected MASTER-NAADP, benzoic acid C-nucleoside, 2'-phospho-3'-adenosine-diphosphate, mimics activity of NAADP in two biological systems, e.g. Ca^{2+} release from permeabilized Jurkat T cells and RYR1 channel opening in lipid planar bilayers (new Fig 3 of revised manuscript). In contrast, the deprotected MASTER-NADP, benzamide C-nucleoside, phospho-adenosine-diphosphate, behaves as an inactive control in these experiments (new Fig 3 of revised manuscript).

The absolute concentrations of MASTER-NAADP required for evoking local or global Ca^{2+} signals depends on (i) diffusion rate of the MASTER-compounds through the plasma membrane, (ii) esterase activity present in the cytosol towards the MASTER-compounds, (iii) degradation of the deprotected NAADP analogue, benzoic acid C-nucleoside, 2'-phospho-3'-adenosine-diphosphate, or of the deprotected NADP analogue, benzamide C-nucleoside, phospho-adenosine diphosphate, and (iv) expression level and activity of xenobiotic efflux pumps. As discussed above (in no.4) these parameters do vary between different cell types and very likely account for the differences observed between T cells, KHYG-1 cells, or Neuro2A cells. However, determination of all these parameters is out of the scope of the current manuscript.

The difference to 2-APB and also to NAADP-AM is that now a control compound, MASTER-NADP, is available to control MASTER-NAADP effects in any cell system, by adding an identical concentration of the control compound. This situation is a huge and significant step forward in the availability of tools

to evoke NAADP signaling, in other words, it is the first reliable tool because MASTER-NAADP control always shows unspecific effects that will result in careful interpretation of results.

7. Figure 4: The authors should mention what imaging speed was used for global calcium imaging. If it is comparable to their calcium microdomain protocol and performed at 40 frames (image pair)/ sec there will be significant bleaching of Fura-2 especially at later time points and could result in underestimation of peaks. If it is not performed at the 40 frames (image pair)/ sec speed and is closer to the usual 1 image pair every 2-4 sec, how do the authors reconcile the loss of several peaks in the intervening period between each image pair with their data in Figure 3?

Reply: The data in new Fig 5 (old Fig 4) are conducted with Fura2-loaded cells at 0.5 ratios/s. This is a standard method used by many researchers to record global Ca^{2+} signaling. The temporal resolution is fairly sufficient to resolve the calcium oscillations, see single tracings below taken from new Fig 5A (left in blue a cell responding with low amplitude; right in red a cell responding with high amplitude). For the high as well as for the low responding cells the global Ca^{2+} oscillations are seen in the range of tens of seconds.

Thus, a loss of a significant number of global Ca^{2+} oscillations is not to be expected under these standard conditions.

8. Figure 4D, why does the mean max amplitude appear very different between panel A and B but quite similar in panel D.

Reply: We are grateful to reviewer 3 for raising this point. Accordingly, we re-analyzed the data set using in new Fig 5 (old Fig 4) the amplitude of the Ca^{2+} peaks (Fig 5C), the number of peaks (Fig 5D), and calculated from these two parameters the mean responsiveness as product of peak number and peak amplitude (Fig 5F). The latter is the most robust way to properly show the effect because both peak number and amplitude are combined.

9. Figure 5: It is unclear how the analysis of NAADP-AM helps in making the central point in the paper and why did the authors not choose 100uM concentration here?

Reply: Currently, NAADP-AM is the only compound available described as a membrane-permeant precursor that liberates NAADP in intact cells. As discussed in detail on p.9/10 of the revised manuscript, synthesis of NAADP-AM using the described method (reference no 28, Parkesh, R. *et al.* Cell-permeant NAADP: a novel chemical tool enabling the study of Ca^{2+} signalling in intact cells. *Cell Calcium* **43**, 531–538 (2008)) always resulted in a mixture of undefined products. This is also true for commercially available NAADP-AM (suppl Figs S9 and S10 of the revised manuscript). Of note, digestion of NAADP-AM by porcine liver esterases did not yield NAADP, making this tool questionable (suppl Figs S9 and S10 of the revised manuscript).

We feel that for the community of NAADP and Ca²⁺ researchers it is important to publish this information in the context of the novel MASTER-NAADP, to clearly see advantages and disadvantages of both compounds or pair of compounds, respectively.

To make a less central point we decided to move the previous Fig 5 into the supplemental part (now suppl Fig S9).

We also used higher concentrations of NAADP-AM (10 and 100 μM, new suppl Fig S8). At 10 μM NAADP-AM does not evoke Ca²⁺ signaling in Jurkat T cells whereas anti-CD3 mAb OKT3 clearly evoked global Ca²⁺ signaling (new suppl Fig S8A). Quantitative analysis of the effects of NAADP-AM at 10 and 100 μM is shown in suppl Fig S8B-E. Neither at 10 μM nor at 100 μM did NAADP-AM addition result in a significant increase over background, as detailed for the mean number of peaks (new suppl Fig S8C) and mean responsiveness as most robust read-out (new suppl Fig S8E).

New suppl Fig S8B-E also demonstrates that such high concentrations of MASTER-NAADP/MASTER-NADP are not suitable for T cells, because unspecific effects of MASTER-NADP increased (compare suppl Fig S8E to Fig 5F). These data again show the huge advantages of the pair of MASTER compounds allowing to custom-tailor the experimental conditions for each cell type.

10. Figure 6: It is intriguing to see that the global response of KHYG-1 and Neuro 2A cells is so different from Jurkat cells in the max amplitude as well as %age of responding cells. Of course the concentration of MASTER-NAADP used is 1000-fold different. However, another possibility is that the levels of Hn1l expression is different across these three cell lines. If the expression of the receptor is indeed 1000-fold different in different cell lines, it needs to be shown by using quantitative analysis of Hn1l levels.

Reply: In the revised version, HN1L/JPT2 expression was analyzed by western blot (new suppl Fig S13). Whereas no quantitative difference was observed for protein expression in the three cell lines Jurkat T cells, KHYG-1 NK cells, and Neuro2A neuronal cells (suppl Fig S13B), slightly different molecular masses were observed (suppl Fig S13A). When compared to recombinant HN1L/JPT2 produced in E.coli (rHN1L/JPT2), HN1L/JPT2 in KHYG-1 cells was of identical molecular mass, whereas slightly elevated molecular masses were observed for Jurkat T cells and Neuro2A cells. The latter indicates different posttranslational modifications of HN1L/JPT2 in different cells types that may influence the differences observed.

Another difference in Figure 4 and 6 is the presence of Synta66 in Figure 4. Why did the authors choose to omit Synta 66 in Figure 6? Wouldn't it be more important in Figure 6 where 100uM of MASTER-NAADP is being used?

Reply: Many more combinations of the pair of MASTER-compounds could have been combined with different pharmacological inhibitors of individual elements of Ca²⁺ signaling. Our intention was more to show specific effects in different cell types to make clear that this pair of MASTER-compounds is widely applicable. Thus, such experiments are certainly interesting, but outside the scope of this manuscript.

11. Figure 3,4,6: Why is the %age of cells responding to MASTER-NAADP only ~15% in Jurkat and KHYG-1? It is ~80% in Neuro2A cells. The authors should resolve this difference in response and perform downstream signaling analysis on a population of cells as suggested below to allow assessment of physiological effects of the pro-drug MASTER-NAADP.

Reply: The percentage of responding Jurkat T cells increased to approx. 90% when 100 μM MASTER-NAADP was used (suppl Fig S8D), quite comparable to the approx. 80% seen in Neuro2A cells with 100 μM MASTER-NAADP (Fig 6J). The lower response rate for Jurkat T cells at 100 nM certainly is explained by the lower concentration of MASTER-NAADP used. For KHYG-1 cells, the low response rate even at 100 μM MASTER-NAADP is difficult to explain. However, human KHYG-1 NK

cells have never been analyzed regarding use of the NAADP signaling pathway; again an interesting point to be investigated in the future, but outside the scope of this manuscript.

General comments:

1. A binding assay between Hn1I and MASTER-NAADP would help further establish the specificity of the MASTER-NAADP.

Reply: We fully agree to reviewer 3. Currently, we are working on such an assay, but we do not think that such pharmacological characterization of the de-esterified MASTER-compounds is essentially required for the current publication.

2. It will strengthen the study if downstream signaling analysis was performed. It will further establish the specificity of MASTER-NAADP.

Reply: Again, we fully agree to reviewer 3 also regarding downstream effects of the pair of MASTER compounds. In T cells, we are currently working on NFAT1 translocation and though this study is not yet finalized, there are initial data showing increased NFAT1 translocation into the nucleus upon MASTER-NAADP addition.

For now, having included not only a highly complex synthesis route to the pair of MASTER compounds, but also the rigorous characterization of these compounds at various concentrations in three different cell lines (plus data in primary T cells), we are confident that the current revised version is of sufficient novelty and significance.

3. Since authors have called MASTER-NAADP a pro-drug, it will be nice to rule out toxicity over prolonged periods. In vivo administration and downstream gene expression analysis would help the study further.

Reply: Again, we grateful to reviewer 3 and thus changed the expression 'pro-drug' to 'precursor' to make clear that any downstream experiments or in vivo administration will be super-interesting aspects to be studied in the future.

Reviewer #4 (Remarks to the Author):

The authors have presented the design, synthesis and testing of a bis-prodrug of benzoic acid modified C-nucleoside 3'-F-adenosine diphosphate for its ability to stimulate NAADP signaling. In that regard, MASTER-NAADP, a Membrane permeAble, STabilized, bioEversibly pRotected prodrug of NAADP was pursued. This process is important as NAADP acts as a second messenger and stimulates Ca²⁺ signaling in many different cell types. The paper is quite well written other than some few typos/grammatical errors below. Interestingly, the chemistry proved to be rather challenging but was ultimately achieved. Moreover, their findings are likely to be of strong interest to the field, due to the development of this useful tool to study NAADP signaling and downstream events. Other than the minor issues below, the paper is ready for publication as it stands.

Reply: We are very glad that, apart from the typos listed below, reviewer 4 states that ' the paper is ready for publication as it stands'.

page 2, line 54: define ER

page 3, line 82: insert "the" before 1,2-position

page 4, line 156: delete also before neuronal

page 5, line 170: insert "a" before few

page 5, line 184: insert "the" before mean

page 5-6: why is figure 5 discussed before figure 4 (i.e. use numerical order)

page 5, line 196: change digest to digestion

page 7, line 251: removed also before "the esterification"

page 10, line 345: should this just say Methods, not online methods?

Reply: All minor points were revised as suggested.

REVIEWERS' COMMENTS

Reviewer #1 (Remarks to the Author):

I recommend the article for publication. By my reading, however, the introduction seems to strawman reference 13. Ref 13 does not invoke nonphysiological pH or nonphysiological concentrations of nicotinic acid and, in fact, explains the regulated production of NAADP that is supported by use of the chemical probes in this paper. Perhaps when the authors make final corrections, they can be more generous to the endolysosomal model of NAADP production.

Reviewer #2 (Remarks to the Author):

The authors have responded satisfactorily to the reviewers' comments. However, there is a discrepancy between the text in section lines 252-264 and Figure S8. The response to reviewer #2 point 4, to reviewer #3 point 9, and the legend of supplementary Figure S8B-E refer to testing two different concentrations of MASTER-NAADP and related compounds in wild type Jurkat cells. The blue bars are 100 μ M and the orange bars are 10 μ M. However, in the manuscript text Figure S8B-E seems to compare wild type and knockout cells. In the discussion of Figure S8B-E, the colors are given as blue for wild type cells and orange for Hn1l/Jpt2^{-/-} cells. The manuscript text in lines 252-264 does not agree with the figure. This confusion needs to be corrected.

Reviewer #3 (Remarks to the Author):

None

Reviewer #4 (Remarks to the Author):

The authors have nicely responded to the comments from the reviewers and the manuscript is ready for publication as it stands.

Reviewer #4 (Remarks on code availability):

Not even remotely my area of expertise, so I cannot comment on the code or its usability.

Point-by-point reply to the remaining comments by the reviewers

REVIEWERS' COMMENTS

Reviewer #1 (Remarks to the Author):

I recommend the article for publication. By my reading, however, the introduction seems to strawman reference 13. Ref 13 does not invoke nonphysiological pH or nonphysiological concentrations of nicotinic acid and, in fact, explains the regulated production of NAADP that is supported by use of the chemical probes in this paper. Perhaps when the authors make final corrections, they can be more generous to the endolysosomal model of NAADP production.

Reply: The text related to NAADP synthesis was modified in a generous way, as asked for by reviewer 1 ('Solid evidence for this endo-lysosomal model of NAADP production was obtained in lymphokine activated killer cells stimulated by interleukin-8¹³').

Reviewer #2 (Remarks to the Author):

The authors have responded satisfactorily to the reviewers' comments. However, there is a discrepancy between the text in section lines 252-264 and Figure S8. The response to reviewer #2 point 4, to reviewer #3 point 9, and the legend of supplementary Figure S8B-E refer to testing two different concentrations of MASTER-NAADP and related compounds in wild type Jurkat cells. The blue bars are 100 μ M and the orange bars are 10 μ M. However, in the manuscript text Figure S8B-E seems to compare wild type and knockout cells. In the discussion of Figure S8B-E, the colors are given as blue for wild type cells and orange for Hn1l/Jpt2^{-/-} cells. The manuscript text in lines 252-264 does not agree with the figure. This confusion needs to be corrected.

Reply: We are grateful to reviewer #2 to raise this point. The text was corrected accordingly.

Reviewer #3 (Remarks to the Author):

None

Reviewer #4 (Remarks to the Author):

The authors have nicely responded to the comments from the reviewers and the manuscript is ready for publication as it stands.

Reviewer #4 (Remarks on code availability):

Not even remotely my area of expertise, so I cannot comment on the code or its usability.